# Federated learning based reference evapotranspiration estimation for distributed crop fields

**Muhammad Tausif**[1], **Muhammad Waseem Iqbal**[1], **Rab Nawaz Bashir**[2,3],
**Bayan AlGhofaily**[3], **Alex Elyassih**[3], **Amjad Rehman Khan**[3]*

**1** Department of Computer Science & Information Technology, The Superior University, Lahore, Punjab, Pakistan, **2** Department of Computer Science, COMSATS University Vehari, Vehari, Punjab, Pakistan, **3** Artificial Intelligence & Data Analytics (AIDA) Lab, College of Computer & Information Sciences (CCIS), Prince Sultan University, Riyadh, Saudi Arabia

* arkhan@psu.edu.sa

**Data Availability Statement:** All Data files are available from the GitHub repository:https://github.com/RaoTausif/FL-Based-ETo-Estimation.

## Abstract

Water resource management and sustainable agriculture rely heavily on accurate Reference Evapotranspiration ($ET_o$). Efforts have been made to simplify the ($ET_o$) estimation using machine learning models. The existing approaches are limited to a single specific area. There is a need for $ET_o$ estimations of multiple locations with diverse weather conditions. The study intends to propose $ET_o$ estimation of multiple locations with distinct weather conditions using a federated learning approach. Traditional centralized approaches require aggregating all data in one place, which can be problematic due to privacy concerns and data transfer limitations. However, federated learning trains models locally and combines the knowledge, resulting in more generalized $ET_o$ estimates across different regions. The three geographical locations of Pakistan, each with diverse weather conditions, are selected to implement the proposed model using the weather data from 2012 to 2022 of the selected three locations. At each selected location, three machine learning models named Random Forest Regressor (RFR), Support Vector Regressor (SVR), and Decision Tree Regressor (DTR), are evaluated for local Evapotranspiration (ET) estimation and the federated global model. The feature importance-based analysis is also performed to assess the impacts of weather parameters on machine learning performance at each selected local location. The evaluation reveals that Random Forest Regressor (RFR) based federated learning outperformed other models with coefficient of determination ($R^2$) = 0.97%, Root Mean Squared Error (RMSE) = 0.44, Mean Absolute Error (MAE) = 0.33 mm day$^{-1}$, and Mean Absolute Percentage Error (MAPE) = 8.18%. The Random Forest Regressor (RFR) performance yields the local machine learning models against each selected site. The analysis results suggest that maximum temperature and wind speed are the most influential factors in Evapotranspiration (ET) predictions.

**Funding:** The research is supported by the Artificial Intelligence & Data Analytics (AIDA) Lab, Prince Sultan University, Riyadh, Saudi Arabia. The authors would like to acknowledge the support of Prince Sultan University, Riyadh Saudi Arabia for support of the Article Processing Charges (APC) of this publication. The funders had a role in study design, data collection and analysis, decision to publish, and preparation of the manuscript.

**Competing interests:** The authors declared that no competing interests exist.

**Abbreviations: ANN**, Artificial Neural Network; **BaTs**, Bayesian Additive Trees; **BoTs**, Boosted Trees; **CNN**, Convolution Neural Network; **DTR**, Decision Tree Regressor; **ELM**, Extreme Learning Machine; **ENR**, Elastic Net Regression; **ET**, Evapotranspiration; **ET$_o$**, Reference Evapotranspiration; **ETR**, Extra Tree Regression; **FFNN**, Feed Forward Neural Network; **FL**, Federated Learning; **GBR**, Gradient Boosting Regression; **IoT**, Internet of Things; **LR**, Lasso Regression; **LSTM**, Long Short-Term Memory; **M5P**, M5 Model Tree; **MAE**, Mean Absolute Error; **MAPE**, Mean Absolute Percentage Error; **ML**, Machine Learning; **MLR**, Multiple Linear Regression; **PCR**, Principal Component Regression; **PLSR**, Partial Least Square Regression; **POR**, Poisson Regression; **PSO**, Particle Swarm Optimization; **R$^2$**, Coefficient of Determination; **RBFNN**, Radial Basis Function Neural Network; **RDR**, Ridge Regression; **RF**, Random Forest; **RFR**, Random Forest Regressor; **RH**, Relative Humidity; **RMSE**, Root Mean Square Error; **SR**, Solar Radiations; **SVR**, Support Vector Regressor; **Tmax**, Maximum Temperature; **Tmin**, Minimum Temperature; **WS**, Wind Speed; **XGBR**, eXtreme Gradient Boosting Regressor.

# 1 Introduction

Water resource management and planning considers Evapotranspiration (ET) as a key water cycle concept and has many application areas [1], i.e., water management, drought monitoring, and irrigation scheduling [2]. Although its computation is critical for water management, the interrelationship between weather parameters and ET makes such computation complex. Moreover, water management in multiple locations is more critical. Therefore, solutions for the ET estimation in multiple locations considering their weather parameters are required to help agriculturists.

In agriculture, farming and irrigation utilize about 70% of fresh water around the globe [3] and significantly impact sustainability [4]. Water management exploits Reference Evapotranspiration ET$_o$ to contribute to improving irrigation practices. Accuracy in ET$_o$ calculation considering the weather condition of a location is compulsory to implement the sustainability practices by irrigation water conservation.

ET$_o$ is a standard unit for calculating the ET rate among locations and weather conditions. It can be computed by estimating a crop's transpired and evaporated water against specific weather conditions [5]. Different weather parameters, i.e., solar radiations, temperature, precipitation, vapor pressure deficit, and wind speed, may influence ET$_o$. ET$_o$ can be estimated using a variety of methods, including Penman-Monteith (PM), Thornthwaite, Hargreaves [6], satellite-based, and machine-learning-based approaches [7–9]. Machine Learning based approaches able to capture more complex relationships between input and output variables [10–12]. Based on weather data, these methods use machine and deep learning algorithms to estimate ET$_o$. Each of these approaches has its advantages and disadvantages. Using historical data, efforts were made to establish empirical relationships between weather parameters and ET$_o$.

Machine learning (ML) algorithms have made remarkable progress in ET estimation with limited weather parameters. ML algorithms, such as Support Vector Regression (SVR), Random Forest Regressor (RFR), and Artificial Neural Networks (ANN), are proven to be very effective for accurate estimation of ET$_o$ with limited weather parameters [13]. ML methods can analyze more complex relationships between the weather parameters for accurate ET$_o$ estimation. Tradition methods for estimating ET$_o$, e.g., the Penman-Monteith equation, are complex and computationally hard. Moreover, simple ML approaches also face limitations, particularly related to data centralization, and may not effectively capture the diverse climatic conditions of different regions.

The application of ML techniques for predicting ET$_o$ leads to evidence of their accuracy by collecting limited weather data. Problems with such measures are that they are often limited to specific areas. The existing solutions were proposed for a specific location tailored according to local weather conditions [14, 15]. The existing solutions of ET$_o$ using limited weather parameters are limited to a specific area, and it is very hard to apply them in different contexts with the same accuracy [16]. The inherent variations in weather conditions of different locations further diversify the situation of ET$_o$ modeling using limited weather parameters [17]. Understanding the relationship between weather and ET$_o$ can be very complex, particularly for several locations with diverse weather patterns [18]. A universal ET$_o$ model using limited weather parameters is difficult to optimize across diverse weather parameters of different locations [17]. Traditional machine learning methods for estimating ET$_o$ are effective within certain contexts but are constrained by their dependence on local datasets, limiting their generalizability across diverse geographical locations. Consequently, this limitation is not due to the models themselves but rather to the restricted scope of the input data available to them. There is a need for a model that can generalize well for ET$_o$ prediction of different locations with distinct weather conditions. This study addresses this limitation by implementing a

federated learning (FL) approach, allowing localized model training and the capability to generalize across these local models using a global model [14]. The proposed approach intends to explore the possibilities of generalizations ability of a federated model to learn $ET_o$ predictions using diverse weather parameters of multiple locations.

FL has the potential to handle different weather conditions across multiple locations using a single global model. Current methods have made some progress in certain areas. However, we still need to figure out how to deal with the diversity of data when estimating $ET_o$ across a wider range of locations using a single model. FL can handle data diversity by leveraging multiple models for different locations [19]. The FL enables distributed data across multiple sources without data centralization [20, 21]. The FL has the potential to address the issues of data heterogeneity, decentralized data distribution, real-time adaptability, and resource efficiency. The advantages of FL over traditional machine learning and deep learning approaches for $ET_o$ estimation of multiple locations with distinct weather conditions make it different, novel, and unique [22]. The architecture of the proposed FL model is illustrated in Fig 1.

FL is diverse in handling the weather parameters of different locations for $ET_o$ estimation considering the given data from multiple locations [23]. It integrates the dataset of multiple locations, improves the accuracy, and generalizes the $ET_o$ estimation model across multiple locations. Furthermore, FL improves geographical coverage and ensures privacy by uncovering the insights of datasets of multiple locations [24, 25]. Considering the mentioned advantages of FL, this paper proposes a novel model to improve the accuracy of $ET_o$ prediction for multiple locations. The proposed model considers the collective power of distributed data [1, 26] and enhances the precision and spatial coverage of $ET_o$ prediction. FL-based $ET_o$ estimation enhances water balance analysis by promoting model generalization and privacy by training on a diverse set of localized data from various regions, ensuring that the models are robust and applicable across different environmental conditions.

The main contributions of the paper are as follows:

- A novel model is proposed for estimating $ET_o$ for three (3) geographical locations in Pakistan by exploiting FL that provides a global automated solution for $ET_o$ prediction and connects agriculturists around the globe.

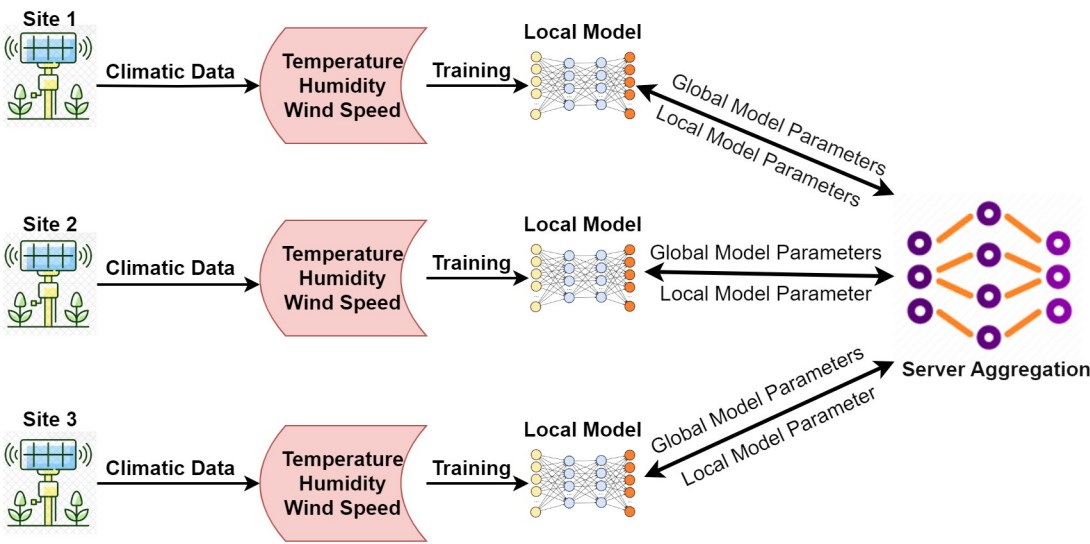

**Fig 1. Overview of the federated model.**

- The study intends to explore the performance of local and global machine learning models in $ET_o$ prediction and their relative comparison.

- The study also explores the impact of different weather parameters in $ET_o$ prediction, using the feature important features of machine learning models.

The remaining sections of the paper are organized as follows: Section II explores existing literature and studies related to $ET_o$ estimation, machine learning models, and relevant methodologies. Section III provides a detailed overview of this study's materials, data sources, and methodologies. Section IV provides the obtained results and discussion, and section V contains a conclusion that summarizes the study's key findings.

## 2 Related work

Machine learning techniques have been the research community's focus in agriculture domain [27–29] in recent years. This section explores the progress of recent emerging approaches for $ET_o$ estimation using machine learning approaches.

Dong et al. [30] proposed a solution that aims to improve the accuracy of $ET_o$ estimations by analyzing the spatial and temporal variation of $ET_o$ in China. This study used three ML models. Included models are; multiple adaptive regression, convolution neural networks (CNN), and extreme learning machines (ELM). CNN provided better estimation results of $ET_o$ estimation.

Rai et al. [31] carried out a comparative analysis of various machine learning models for the prediction of monthly $ET_o$. They used India's weather data for a period from 2009 to 2016. The results revealed that among all the models examined, the SVR model yielded the highest accuracy in reconstructing water requirements.

Bellido et al. [32] discussed the application of the neural network for determining the $ET_o$ in the Andalusia region of southern Spain. The multi-layer perceptron, ELM, SVM, generalized regression neural network (GRNN), RF, and XGBoost were the methods assessed in this study. This study employed performance measures such as the coefficient of determination (R2), the root mean squared error (RMSE), and Nash-Sutcliffe model efficiency coefficient (NSE). Notably, the ELM approach emerged the best of all models, with an R2 of 0.89 NSE, 0.89, and RMSE of 0.67mm day$^{-1}$.

Krishna et al. [33] have employed diverse cognitive computing models to predict $ET_o$. The study utilized various factors and concluded that the second order neural network was most accurate in predicting $ET_o$. It also showed low error and high accuracy with the use of RMSE and R2 values of 0.065 mm day$^{-1}$ and 99%, respectively.

Ayaz et al. [34] used different machine learning models in India and New Zealand. The focus of this study is to use just only temperature data. They tried models like Long Short-Term Memory (LSTM), XGBR, SVR, and RF. When using all weather data inputs, the LSTM model outperformed with 99% accuracy. But when they only used temperature, accuracy dropped to 86%.

Samman et al. [35] examined the performance of four machine learning models in $ET_o$ estimations. Five Iraqi stations were used as inputs to the models. SVM, RF, Bagged Trees (BaTs), and Boosting Trees (BoTs) have all been used for modeling daily $ET_o$. The RFR model provided the most accurate $ET_o$ estimates at all cites, while SVM provided the lowest results. RFR significantly enhanced estimation accuracy compared to SVM, BoT, and BaT models across different locations. The improvement in RMSE ranged from 8% to 94% during the test period.

Mirzania et al. [36] proposed $ET_o$ estimation approach for Australia. This study evaluated the performance of three models: the innovative Gunner algorithm, SVR, and hybrid

innovative Gunner support vector regression. It was found that the AIG-SVR model outperformed, with r and RMSE corresponding to Marree Aero station values of 0.945 and 1.124, respectively, and St Helen Aerodrome stations of 0.951 and 0.476, respectively.

Khan et al. [16] proposed a method for reclaiming saline soil that employs the Internet of Things (IoT) and ML to estimate $ET_o$ on monthly data. LSTM and ensemble LSTM models predict ETs based on field temperature, irrigation water salinity, and soil salinity. It was found that the ensemble LSTM-based model was more accurate than the single LSTM model, with an accuracy of 92% for the $ET_o$ estimation.

Rashid et al. [37] aimed to develop an $ET_o$ estimation method using four machine learning models with different input combinations. All combinations of the four defined models showed that the RF model was the most effective with MAE, $R^2$ and RMSE values 0.76 mm $day^{-1}$, 0.85% of 0.82 mm $day^{-1}$ respectively.

Yu et al. [38] aimed to assess the performance of different machine learning models for $ET_o$ estimation with various input combinations, such as minimum and maximum temperatures, wind speed, solar radiation, relative humidity, atmospheric pressure, and sunshine duration. This study evaluated the performance of three machine learning models: ANN, SVR, and ELM. The SVR model proved to be the most accurate, with an R of 0.881, an RMSE of 0.925mm $day^{-1}$, MAE of 0.59 mm $day^{-1}$, and NSE of 0.744.

Zhang et al. [39] provide a detailed analysis of the special specificity of FL and the potential future of FL. FL is important in numerous contexts of application, and in particular when it is used for the discussion of frameworks of IoT. The research also addresses the challenges of applying FL within the IoT framework. The work also outlines practical aspects concerning the implementation of FL in practice and the necessity of the corresponding development tools.

Using a federated approach, Manoj et al. [40] introduced crop yield prediction on distributed datasets across multiple client devices. The ResNet-16 and ResNet-28 regression models were trained with the "federated averaging" technique to ensure decentralized training. The results of these models were then compared with other deep learning, and machine learning models. This research indicates that federated averaging was effective when applied with the ResNet-16 regression model and Adam optimizer to enhance the performance.

Kumar et al. [41] address the issues of data privacy and security that affect the implementation of SA. The study proposes PEFL: private FL framework with distinctive depth of privacy encoding. To enhance privacy, PEFL employs perturbation based encoding while the short term memory-auto encoder enhances the capacity of the memory. Despite a somewhat ambiguous division between the standard and the attack pattern, PEFL outperforms the non-FL and other FL methods for the ToN-IoT dataset.

Nguyen et al. [42] focused on the growing popularity of FL in IoT networks. The overview of FL and IoT with the overview of the improvements made in the recent advancements is presented. This study also examines FL possibility of enabling a range of IoT services, including data sharing, caching, attack detection, localization, mobile crowdsensing, and privacy protection. FL is extensively analyzed across critical IoT domains such as healthcare, transportation, UAVs, intelligent cities, and industry, highlighting its transformative impact.

Imteaj et al. [43] conducted a study that examined how distributed machine learning models could be trained on IoT devices with limited resources. It describes the prior research on FL and the assumptions made about its widespread use through IoT devices. The study also discussed the difficulties and problems of integrating FL into an IoT environment. A thorough analysis of new obstacles to using FL in diverse IoT scenarios is presented. In the estimation of advancing $ET_o$ recent studies have contributed valuable understandings. Combining remote sensing techniques lays the foundation for satellite data to understand river dynamics [44].

**Table 1. Comparison of the state-of-the-art approaches.**

| Researcher(s) | Dataset | Methodology | Performance |
|---|---|---|---|
| Zhu et al. (2020) [46] | China | PSO-Extreme Learning Machine | $R^2$: 0.986, RMSE: 0.23 mm day$^{-1}$ |
| Gong et al. (2021) [47] | China | Cubist Regression, Support Vector Regression | $R^2$: 0.995, RMSE: 0.26 mm day$^{-1}$ |
| Duhan et al. (2023) [48] | India | Least Squares SVM | $R^2$: 0.998, RMSE: 0.73 mm day$^{-1}$ |
| Aly et al. (2023) [49] | Egypt | SVR, ETR, KNN, Adaboost regression | $R^2$: 0.925-0.990, RMSE: 0.55-0.99 mm day$^{-1}$ |
| Nagappan et al. (2020) [50] | India | Deep learning neural network | $R^2$: 0.970, RMSE: 0.17 mm day$^{-1}$ |
| Dias et al. (2021) [51] | Brazil | MLR, RF, PCR, GBR | $R^2$: 0.910, NSE: 0.90 mm day$^{-1}$ |
| Mokari et al. (2022) [52] | Mexico | ELM, SVR, GP, SVR | MAE: <0.5 mm day$^{-1}$, RMSE: <0.5 mm day$^{-1}$ |
| Reis et al. (2019) [53] | Brazil | ANN, ELM, MLR | R: 0.718, RRMSE: 0.165 mm day$^{-1}$ |
| Elbeltagi et al. (2023) [54] | Egypt | REPTree model | R: 0.991, RMSE: 0.37 mm day$^{-1}$ |
| Rajput et al. (2023) [55] | India | MLR, ENR, RDR, PLSR, POR, LR | $R^2$: 0.994, RMSE: 0.136 mm day$^{-1}$ |
| Santos et al. (2023) [56] | Minas Gerais state | ANN, RF, SVM, MLR | $R^2$: 0.976, RMSE: 0.168 mm day$^{-1}$ |
| Estevez et al. (2020) [57] | Spain | Wavelet analysis with ANN | R: 0.760-0.900, RMSE: 679.-29.82 mm day$^{-1}$ |
| Achite et al. (2022) [58] | Algeria | FFNN, RBFNN, and GEP | $R^2$: 0.992, RMSE: 0.234 mm day$^{-1}$ |
| Dong et al. (2022) [30] | China | CNN, ELM | $R^2$: 0.992, RMSE: 0.234 mm day$^{-1}$ |
| Rai et al. (2022) [31] | India | SVR, M5P, RF | MAE: 0.076, RMSE: 0.110 mm day$^{-1}$ |
| Karishna et al. (2021) [33] | India | Second order neural network | $R^2$: 0.996, RMSE: 0.065 mm day$^{-1}$ |
| Bellido et al. (2021) [32] | Southern Spain | MLP, ELM, SVM | $R^2$: 0.890, RMSE: 0.670 mm day$^{-1}$ |
| Ayaz et al. (2022) [34] | India, New Zealand | LSTM, XGBR, SVR, RF | $R^2$: 0.996, RMSE: 0.650 mm day$^{-1}$ |
| Samman et al. (2023) [35] | Iraq | SVM, RF, BaTs, BoTs | $R^2$: 0.992, RMSE: 0.077 mm day$^{-1}$ |
| Mirzania et al. (2023) [36] | Australia | Gunner algorithm, SVR, AIG-SVR | R: 0.945, RMSE: 1.124 mm day$^{-1}$ |
| Khan et al. (2022) [16] | Pakistan | LSTM, ensemble LSTM | $R^2$: 0.920, Pearson correlation: 0.916 |
| Rashid et al. (2021) [37] | North Dakota | SVM, LR, RF, GEP | $R^2$: 0.850, RMSE: 0.760 mm day$^{-1}$ |
| Yu et al. (2020) [38] | China | ANN, SVR, ELM | R: 0.973, RMSE: 0.118 mm day$^{-1}$ |

[45] explores the potential of machine learning and satellite data for enhancing seasonal water supply forecasts. These studies collectively underscore the multidisciplinary approach required to refine reference evapotranspiration modeling and emphasize the importance of remote sensing data in $ET_o$ estimation.

Although significant improvements in existing approaches (as shown in Table 1) have been observed, these approaches still suffer from limited geographical coverage, highlighting the need to address data diversity in $ET_o$ estimation across a broader range. Despite significant advancements in the estimation of $ET_o$ through various traditional and machine learning methods, a notable gap exists in the literature concerning the application of these approaches across multiple geographical locations with diverse weather conditions. Most existing studies are limited to localized datasets, effectively modeling $ET_o$ for specific areas. The existing models are limited in generalizing findings across different locations with diverse weather parameters.

For instance, while numerous researchers have successfully applied machine learning techniques such as Support Vector Regression (SVR) and Random Forest Regressor (RFR) to predict $ET_o$ in singular climatic contexts, the results are often not transferable to other regions with different meteorological characteristics. This limitation is primarily due to the inherent variability in weather parameters—such as temperature, humidity, wind speed, and solar radiation—that influence $ET_o$ differently in distinct environments.

There is a needed to integrate FL techniques to estimate $ET_o$ across multiple locations simultaneously. By using an FL framework, this study aims to use localized data from diverse climatic conditions while ensuring model generalization. This approach not only addresses the

existing limitations in the literature but also provides a comprehensive solution for improving $ET_o$ estimations relevant to agricultural practices and water resource management across different geographical settings.

## 3 Material and methods

This section details the key components and methodologies used in this $ET_o$ estimation study based on ML algorithms and FL. The PM equation has been recognized as a standard [59]. However, it gets complicated because it needs many different factors to operate. The PM equation is written as (1)

$$ET_o = \frac{0.407\lambda\Delta(R_n - G) + \gamma(8910T_n - T_a)WS(e_s - e_a)}{\Delta + \gamma(1 + 0.34WS)} \tag{1}$$

Where,

| | | |
|---|---|---|
| $ET_o$ | Reference Evapotranspiration | (mm day$^{-1}$) |
| $R_n$ | Net radiation at the crop surface | (MJ m$^{-2}$ day$^{-1}$) |
| $G$ | Soil heat flux density | (MJ m$^{-2}$ day$^{-1}$) |
| $T_n$ | Mean air temperature | ($^{\circ}$C) |
| $WS$ | Wind speed | (m s$^{-1}$) |
| $e_s$ | Saturation vapor pressure | ($^{\circ}$C) |
| $e_a$ | Actual vapor pressure | ($^{\circ}$C) |
| $\Delta$ | Slope vapor pressure curve | (kPa $^{\circ}$C$^{-1}$) |
| $\gamma$ | Psychrometric constant | (kPa $^{\circ}$C$^{-1}$) |
| $\lambda$ | Latent heat of vaporization of water | (MJ kg$^{-1}$) |

A reliable $ET_o$ estimate is essential for water resource management, agriculture, and weather sustainability. Using FL, we address the challenges associated with aggregating data from diverse geographic regions while maintaining the model's generalizability and the data's privacy.

### 3.1 Study area

Pakistan's diverse climate and geography result in varying ET rates across different regions [60–64]. The study area comprises Punjab, Pakistan's second-largest province in the eastern part of the country, as shown in Fig 2.

Agriculture is an important sector of Pakistan's economy. This sector directly supports the country's population and accounts for 26 percent of gross domestic product [65]. The major crops include cotton, wheat, rice, sugarcane, fruits, and vegetables. Multan, Faisalabad, and Rawalpindi are three major cities in Pakistan known for their fertile lands and significant contributions to the Pakistan agriculture sector. Geographically, Punjab is located between 24–37˚ N and 62–75˚ E. The majority of Punjab falls within the arid and semi-arid zones. Punjab, Pakistan, can be categorized into three distinct regions. The Multan region is characterized by arid conditions and experiences relatively high temperatures. The climate here is notably harsh. The Faisalabad area is semi-arid. The climate in Faisalabad is generally milder than that in the southern region. Rawalpindi features a tropical and semi-arid climate. Consequently, it typically experiences high temperatures. These weather distinctions within Punjab are essential

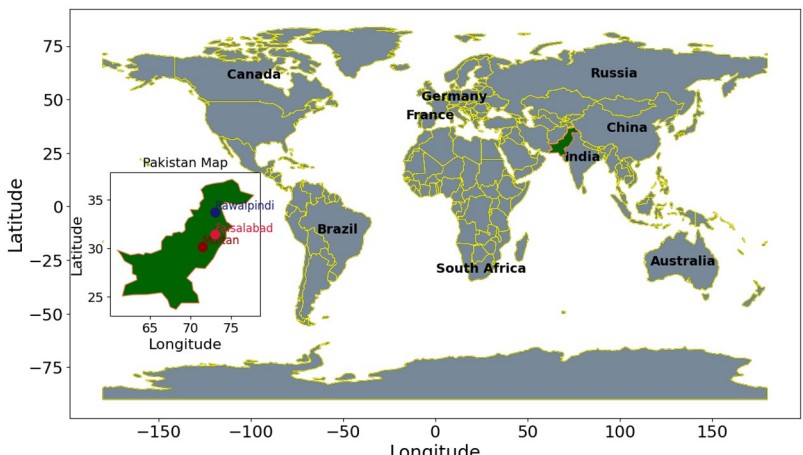

**Fig 2. Geographical locations of the experiment cities.**

when studying various weather and agricultural phenomena in the region. These distinctions significantly impact factors such as $ET_o$ and water resource management.

### 3.2 Dataset

The data for this study were collected from three stations in Punjab, Pakistan. Multan is located at 30.1575˚ N, 71.5249˚ E, Rawalpindi is at 33.5651˚ N, 73.0169˚ E, and Faisalabad is at 31.4187˚ N, 73.0791˚ E. Daily data for 2012–2022 was obtained from NASA data sources [66] and Panman moniteth equation is used to calculate the daily $ET_o$. These selected stations cover the southern, central, and upper parts of Punjab. Daily data were collected on three key parameters: Maximum temperature (Tmax), wind speed (WS), and relative humidity (RH). The weather in these areas is distinct, with significant differences in weather characteristics. Figs 3–5 show separate plots for three distinct datasets. These 3D-scatter plots visualize the relationships between selected features Tmax, WS, RH, and $ET_o$ Within each loop iteration. Adding climatic conditions into traditional ML models is helpful but doesn't address challenges like data centralization, regional variability, and privacy. FL making it ideal for $ET_o$ estimation in distributed crop fields.

The dataset displays diverse weather characteristics at Multan, Faisalabad, and Rawalpindi (presented in Table 2). The temperatures in Multan and Faisalabad are higher, with Multan recording the highest Tmax at 50˚C and Faisalabad following closely at 49.5˚C. Rawalpindi, on the other hand, is comparatively milder, with a Tmax of 47.4˚C. Rawalpindi shows the maximum RH of 94.5% and Multan the lowest minimum RH of 7.3%. Significant variations in WS in different cities, with Faisalabad having the highest maximum WS 5.67 ms$^{-1}$. The $ET_o$ rate varies greatly in Multan, indicating substantial water loss. However, the rate is lower in Faisalabad and Rawalpindi due to their relatively milder climates. These observations underscore the diverse meteorological conditions among the regions.

The diagram presented in Fig 6 consists of a set of violin plots describing the data distribution of five climatic variables. It shows the probability density of the data at different values, with the plot's width representing the density and a central box plot indicating the interquartile range, median, and potential outliers. Each subplot is labeled from A to E, with each violin plot showing the data distribution. Faisalabad and Multan show a wider spread of Tmax values from around 10˚C to 50˚C while mean temperatures range approximately from 10˚C to 40˚C.

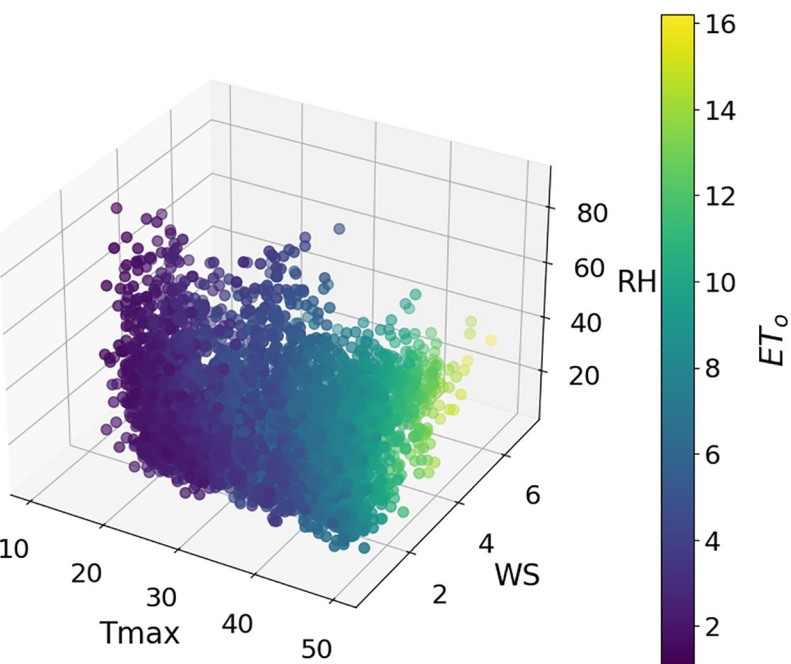

**Fig 3. Relationship of the selected features with ET$_o$ at Multan dataset.**

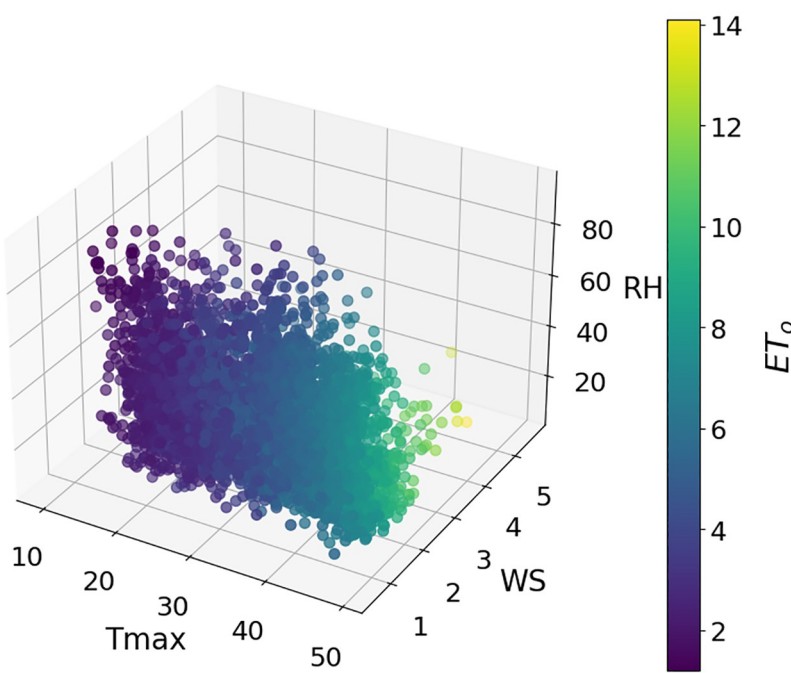

**Fig 4. Relationship of the selected features with ET$_o$ at Faisalabad dataset.**

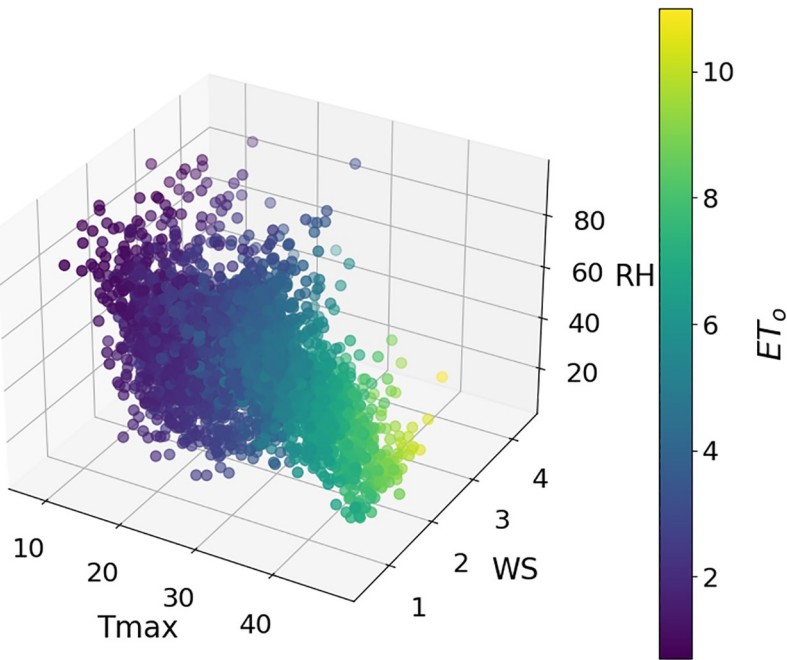

**Fig 5. Relationship of the selected features with $ET_o$ at Rawalpindi dataset.**

Faisalabad and Multan have similar distributions of wind speeds reaching up to 7 m/s. Faisalabad and Multan display a broader range of evapotranspiration values, with distributions extending up to around 15 mm/day. Rawalpindi's $ET_o$ values show fewer variations in evapotranspiration rates. These values are significant for understanding regional climatic conditions and their agricultural implications.

Scatter plots are generated to allow comparison and visualization of data characteristics across datasets, as shown in Figs 7–9. Figs 7–9 used in this study to illustrate how these datasets are associated with estimates of $ET_o$. This also illustrates the different statistical properties of each dataset, enabling a clear understanding of their characteristics. Variations in data

**Table 2. Summary statistics of weather variables in Multan, Faisalabad, and Rawalpindi.**

| Dataset | Variable | Count | Mean | SD | Min | Max | Median |
|---|---|---|---|---|---|---|---|
| Multan | Tmax | 4018 | 34.94 | 8.53 | 10.2 | 50 | 36.7 |
| | WS | 4018 | 1.96 | 1.00 | 0.38 | 7.07 | 1.66 |
| | RH | 4018 | 34.99 | 14.90 | 7.3 | 88.6 | 32.5 |
| | $ET_o$ | 4018 | 5.76 | 2.82 | 1.1 | 16.2 | 5.3 |
| Faisalabad | Tmax | 4018 | 33.33 | 8.36 | 9.2 | 49.5 | 34.9 |
| | WS | 4018 | 1.52 | 0.56 | 0.52 | 5.67 | 1.42 |
| | RH | 4018 | 40.19 | 15.90 | 6.3 | 93.6 | 38.35 |
| | $ET_o$ | 4018 | 5.14 | 1.98 | 1.2 | 14.1 | 4.95 |
| Rawalpindi | Tmax | 4018 | 28.69 | 7.67 | 7.8 | 47.4 | 29.6 |
| | WS | 4018 | 1.49 | 0.44 | 0.49 | 4.37 | 1.42 |
| | RH | 4018 | 52.77 | 17.24 | 7.9 | 94.5 | 52.65 |
| | $ET_o$ | 4018 | 3.99 | 1.88 | 0.7 | 11 | 3.6 |

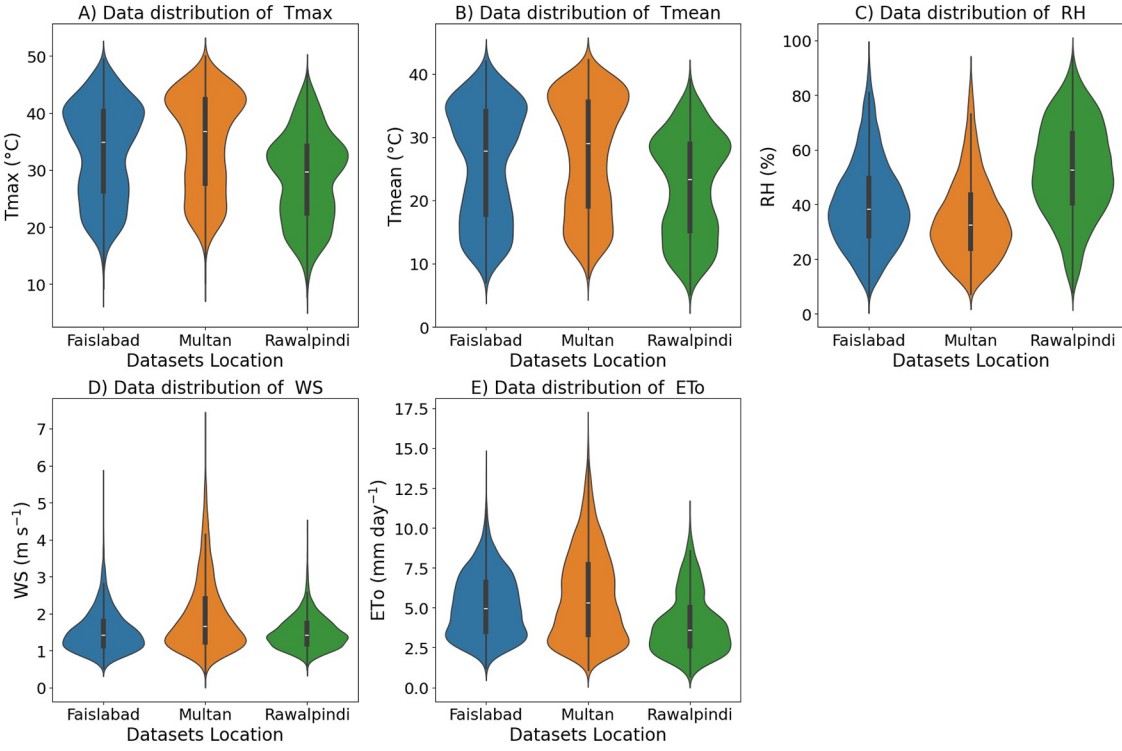

**Fig 6. Violin plots for each variable in each dataset.**

distributions could significantly impact FL models' ability to generalize and predict $ET_o$. $ET_o$ is more accurately simulated by including Tmax, wind speed, and humidity as inputs to the model. The Tmax, WS and RH strongly correlate with $ET_o$. These factors directly affect $ET_o$ as they govern the rate at which water converts from liquid to vapor. The inclusion of these parameters in $ET_o$ models allows better capture of the complex dynamics of water loss processes.

The correlation plots are shown in Figs 10–12. Correlation plots visually represent the relationships between variables in a dataset, often using color-coded matrices. These plots display the strength and direction of correlations, with the intensity of color or the slope of the trend line indicating the degree of positive or negative correlation between pairs of variables. In our study, these correlation plots show the relationships between $ET_o$ and feature sets. It is evident from the correlation results that there is a strong relationship between weather parameters and $ET_o$. Fig 13 shows weather parameters across different datasets over the years.

### 3.3 Machine learning models

**3.3.1 Decision Tree Regressor (DTR).** DTR is a type of decision tree used in supervised machine learning tasks like regression and classification. The working principle is making a tree of decisions based on various features in the dataset, such as temperature, wind speed or humidity. DTR divides the data into smaller groups until it cannot be divided anymore or the stopping criteria are met. DTR accommodates various data types and is adept at capturing complex non-linear connections within the dataset.

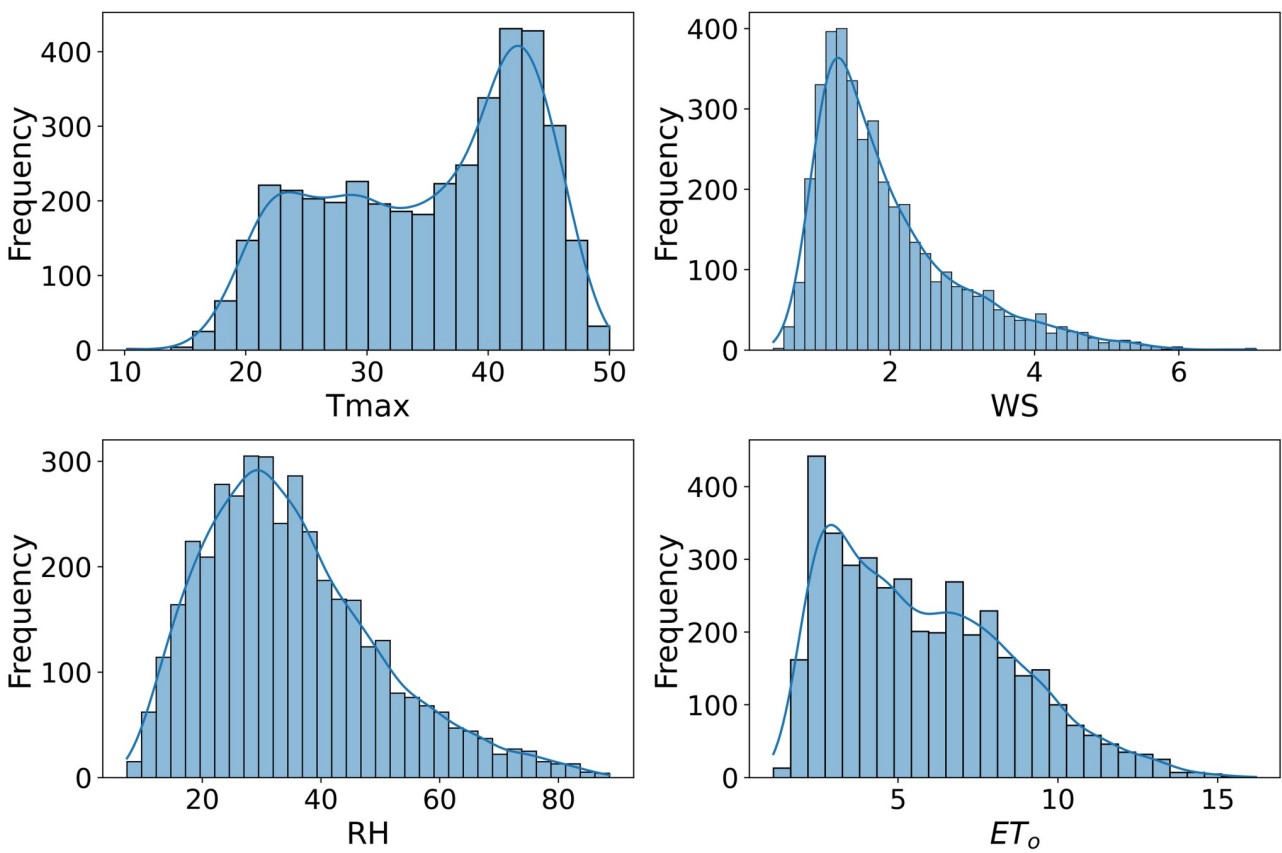

**Fig 7. Distribution plot of Multan dataset.**

**3.3.2 Random Forest Regressor (RFR).** RFR used an ensemble learning approach and acted like a team of DTR models working together to enhance the prediction of the task. It combines many tree models to give a final prediction by averaging what each model (tree) predicts. Each tree in the prediction is trained on a random subset of the training data. For each tree split, a random subset of features is considered. This ensemble approach helps to prevent the model from memorizing the training data, which could lead to wrong predictions on new data. RFR Can handle large datasets with high dimensionality.

**3.3.3 Support Vector Regressor (SVR).** SVR is a supervised machine learning tool used to perform regression tasks. SVR works by finding a best-fit line (hyperplane) to the data, leaving some points outside but not too many. SVR focuses on controlling how many points can be outside this line rather than trying to make every prediction perfect. SVR is Memory efficient, using only support vectors in the decision function. Using different kernel functions for non-linear decision boundaries makes it more efficient to perform regression tasks.

## 3.4 Feature importance analysis

Feature importance analysis determines the impact and importance of different weather parameters for other locations on a specific location's $ET_o$. Gini impurity metric is used to assess the importance of various weather parameters in predicting $ET_o$ of each location. Gini impurity is a commonly used criterion in decision tree algorithms to evaluate the quality of a

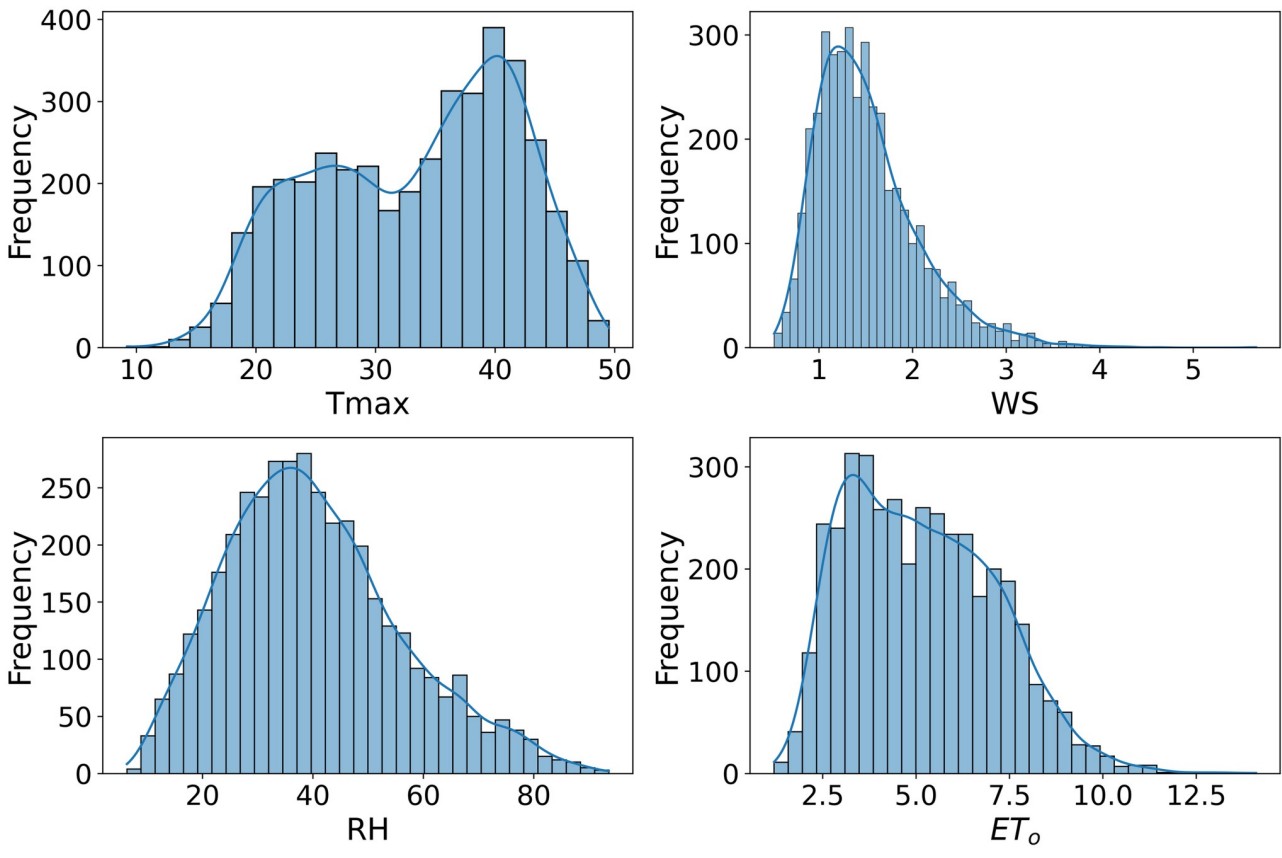

**Fig 8. Distribution plot of Faisalabad dataset.**

split at each node. The Gini impurity for a dataset is calculated using the Eq 1.

$$Gini = 1 - \sum_{i=1}^{n} (p_i)^2 \qquad (2)$$

Where:

- $p_i$ is the proportion of instances in class $i$ relative to the total number of instances.

- $n$ is the different values of $ET_o$.

The Gini impurity is calculated for the parent and child nodes after the split when a feature is used to split the data at a node. The decrease in Gini impurity resulting from this split indicates how well the feature separates the data into different $ET_o$ value ranges. The more significant the reduction in impurity, the more important the weather feature is considered for $ET_o$ determination. The total decrease in Gini impurity is calculated for each feature in the model, which is determined by splits on that feature across all trees in the Random Forest model. The total decline is normalized by the number of trees to determine the average Gini importance score for each feature.

The resulting scores indicate the relative importance of each weather parameter in predicting $ET_o$. A higher Gini importance score suggests a feature significantly impacts the $ET_o$ predictions.

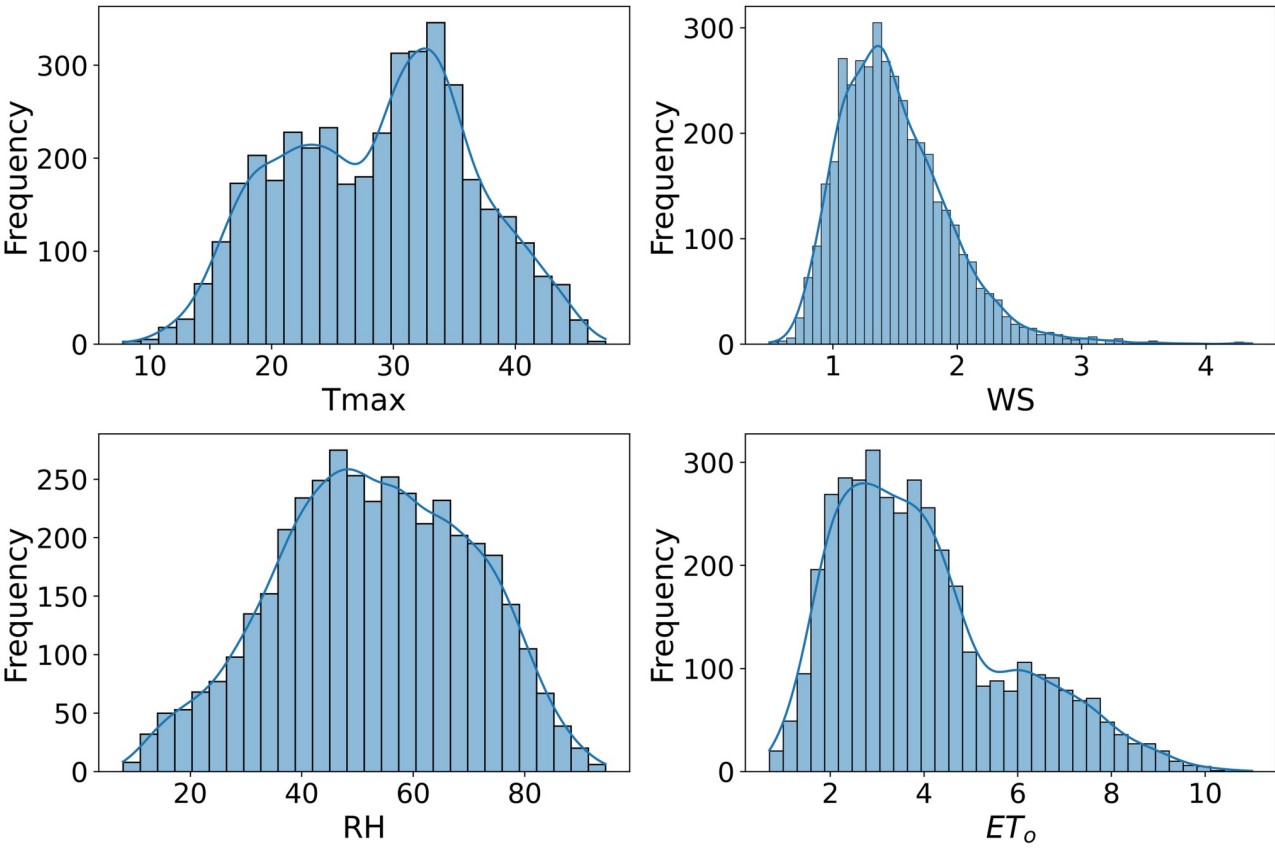

**Fig 9. Distribution plot of Rawalpindi dataset.**

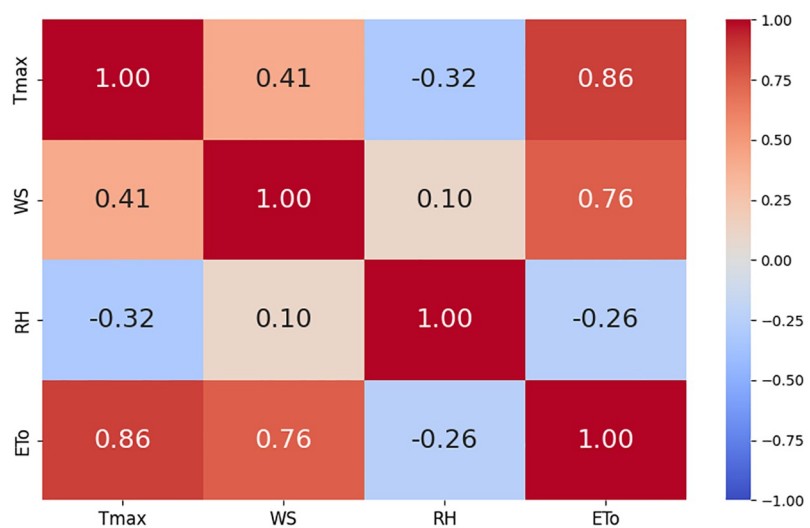

**Fig 10. Correlation among climatic variables at Multan dataset.**

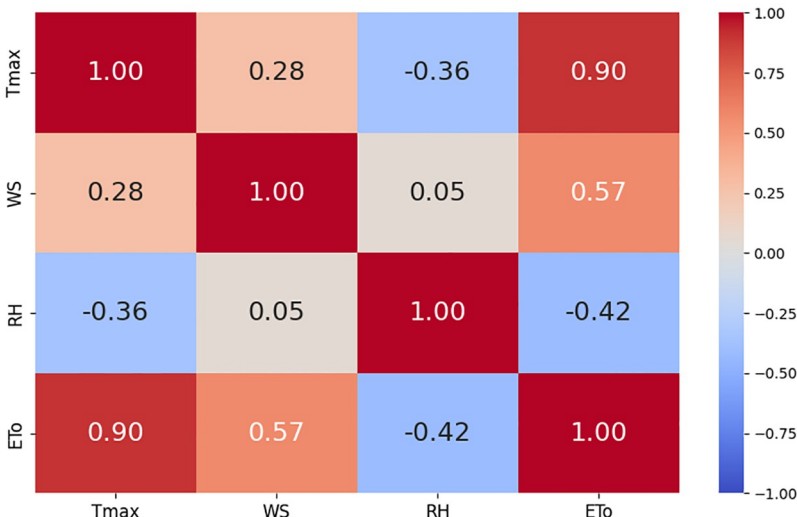

**Fig 11. Correlation among climatic variables at Faisalabad dataset.**

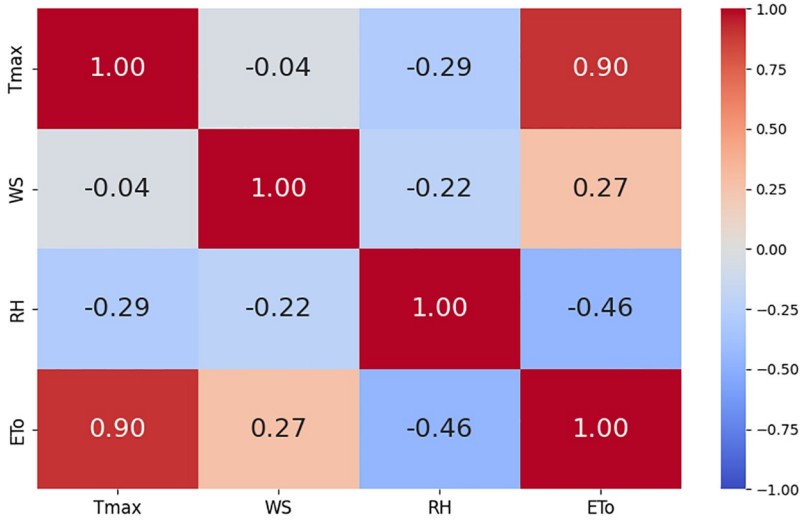

**Fig 12. Correlation among climatic variables at Rawalpindi dataset.**

The Gini impurity criterion is used to assess the importance of weather parameters for $ET_o$ estimation because it can handle continuous variables without considering data distribution. Gini impurity is also computationally efficient and allows seamless integration with RF models to apply an ensemble learning approach to enhance the predictive accuracy of the model.

### 3.5 Federated Learning (FL) framework for $ET_o$ estimation

The proposed FL framework adopts a decentralized architecture, enabling multiple clients to collaboratively train a global model for estimating evapotranspiration ($ET_o$). Each client is responsible for training on its local dataset, which contains weather parameters relevant to $ET_o$ estimation, as shown in Fig 14. The central server orchestrates the training process by

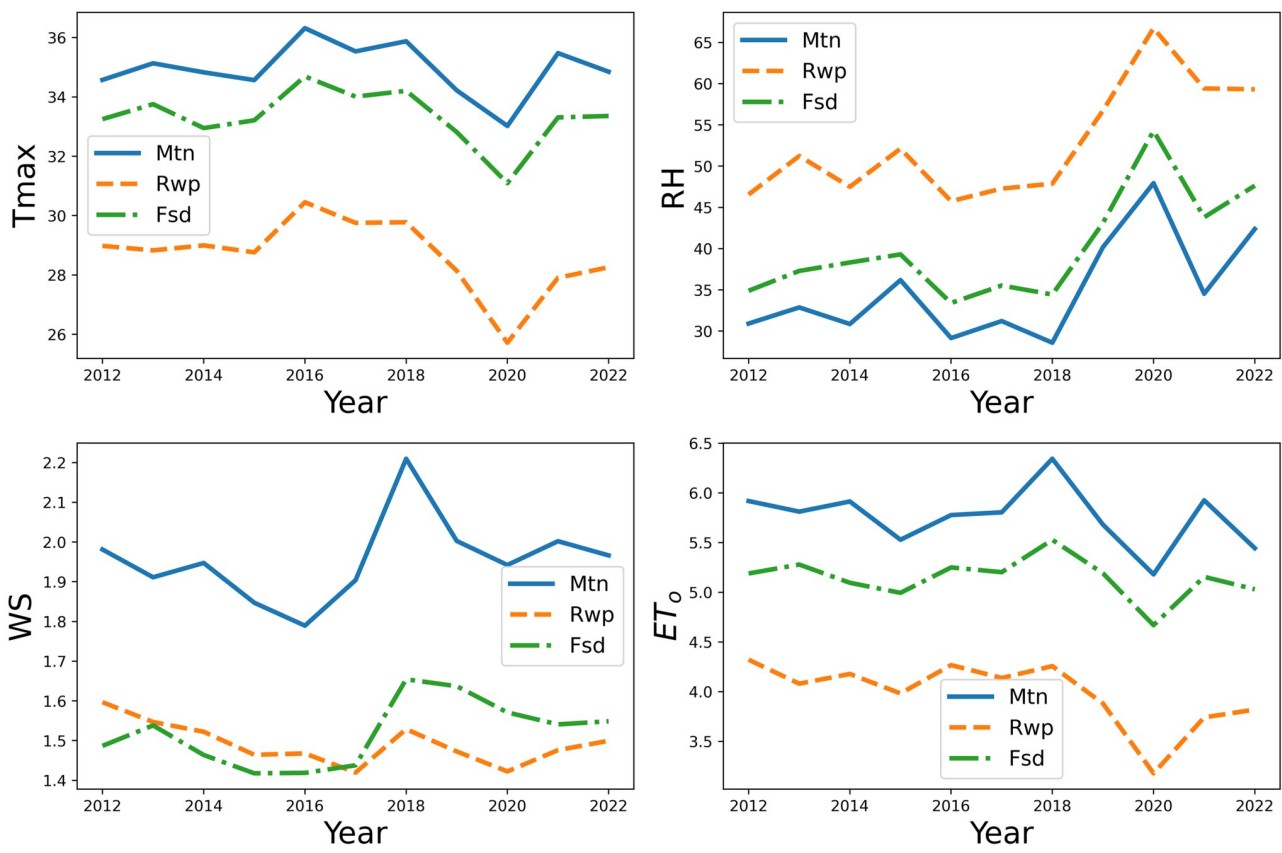

**Fig 13. Comparison of input variables and ET$_o$ over the years at all sites.**

coordinating model updates across clients. This collaborative approach is particularly advantageous for estimating ET$_o$ in distributed crop fields, where local weather data varies, and direct data centralization may not be feasible due to privacy, bandwidth, or regulatory concerns.

**3.5.1 FL framework design and methodology.** The core of the framework involves three key components: client initialization, local training, and global aggregation. The workflow of the federated learning process applied to ET$_o$ estimation is as follows:

1. **Client Initialization:** Each client initializes its local model by receiving parameters from the global model, which is maintained by the central server. The local model represents an initial estimate for ET$_o$ based on the global understanding of weather patterns.

2. **Local Training:** Clients train their local models using their respective datasets, which include historical weather data such as maximum temperature, wind speed, and relative humidity collected over the period from 2012 to 2022. Each client optimizes its local model parameters based on a loss function defined specifically for ET$_o$ prediction. The loss function typically measures the error between predicted and actual ET$_o$ values at the local level, helping each client refine its model.

3. **Model Update:** After local training, clients compute updates to their model parameters. These updates are derived from the gradient of the loss function with respect to the local model parameters, representing the direction and magnitude of adjustments needed to

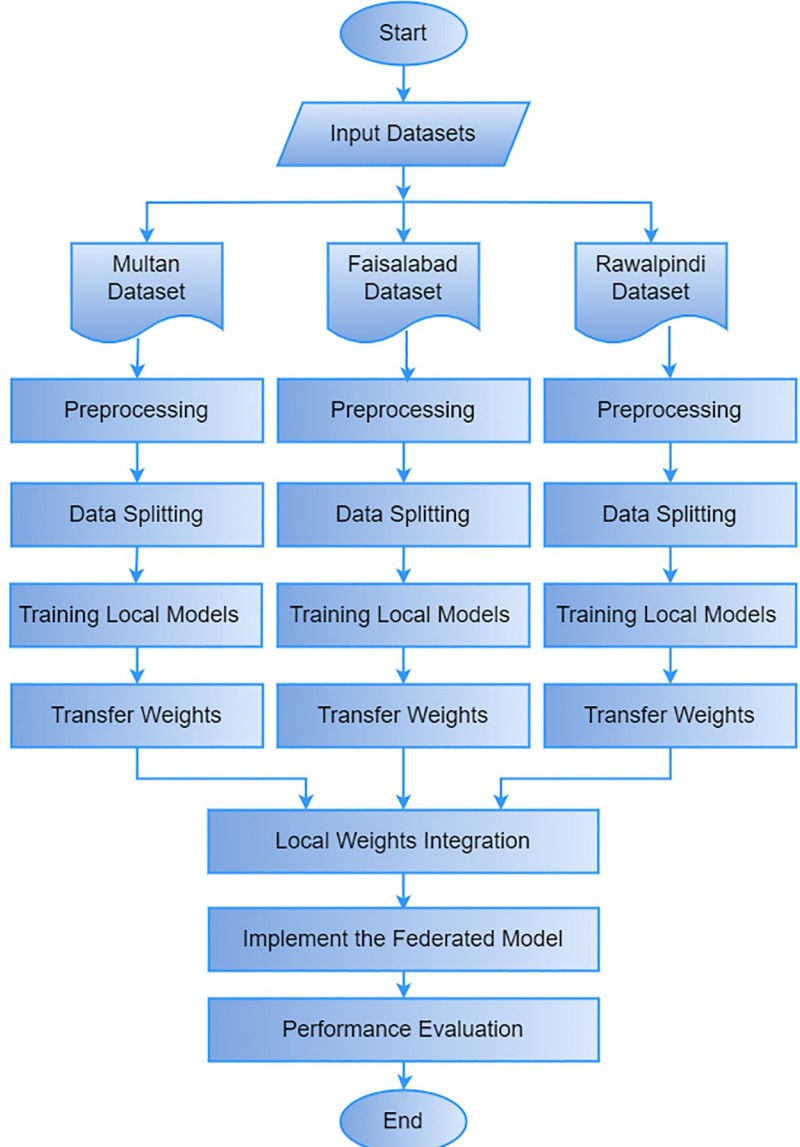

**Fig 14. FL-based ET$_o$ prediction flow chart.**

improve the model's accuracy. Clients then send these updates to the central server for aggregation.

4. **Aggregation:** The server aggregates the updates from all clients to form a new global model, which incorporates the knowledge from all participating regions. The aggregation process involves averaging the model parameter updates from each client. This ensures that the global model reflects both the local weather conditions (which vary by region) and the generalizable patterns across all locations. The typical optimization objective in FL can be expressed as:

$$F(\theta) = \frac{1}{n} \sum_{i=1}^{n} L_i(\theta), \qquad (3)$$

where $F(\theta)$ is the global loss function to be minimized, representing the overall model performance across all clients, $\theta$ represents the global model parameters, $n$ is the total number of participating clients, and $L_i(\theta)$ is the local loss function for client $i$, which is computed using the client's local dataset and global model parameters $\theta$.

The goal of FL is to minimize the average of these local losses across all clients, ensuring that the global model performs well across diverse weather conditions.

The process of aggregation is mathematically represented by the following equation for the global model update:

$$\Delta\theta_{\text{global}} = \frac{1}{n}\sum_{i=1}^{n}\Delta\theta_i, \tag{4}$$

where $n$ is the number of participating clients, $\Delta\theta_i$ is the update computed by client $i$ based on its local training, and $\Delta\theta_{\text{global}}$ represents the aggregated update applied to the global model.

This aggregation ensures that the final model is a synthesis of all local models, enhancing the model's ability to generalize across different climates and geographical regions.

5. **Clients and Communication Process:** In this study, three clients represent three distinct geographical locations: Multan, Faisalabad, and Rawalpindi. Each client collects weather data over the period from 2012 to 2022, focusing on parameters like maximum temperature, wind speed, and relative humidity. The communication process is designed to minimize the need for large-scale data transfer and ensure that local data privacy is maintained. The communication process includes the following steps:

   (a) **Data Characteristics:** Each client's dataset is unique, reflecting local weather conditions and variations in $\text{ET}_o$ across the regions. This diversity in the data is essential for training a robust and generalized model that can adapt to different climates.

   (b) **Communication Rounds:** The FL process consists of 20 communication rounds. In each round, the central server distributes the updated global model to all clients and receives their local model updates. This iterative process continues until convergence is achieved, meaning that the global model performs satisfactorily across all regions.

The Algorithm regarding FL training is given by Algorithm 1. The FL algorithm starts with **Initialization** (lines 2-4), where the global model parameters ($w$) are set. The learning rate ($\alpha = 0.025$), regularization strength ($\lambda = 0.05$), and number of training rounds ($T = 20$) are defined. The optimal values are found using the random search method. During the **Training Process** (lines 5-11), for each training round $t$ (line 5), each client $i$ (line 6) updates its local model parameters based on the global model $w$ and its local dataset ($X_i$, $Y_i$). This update involves adjusting the local model parameters according to the gradients from the local loss function and regularization term. After all clients have completed their local updates, the **Global Model Update** (line 8) takes place, where the global model parameters are updated by averaging the parameters from all clients. Finally, the **Convergence Check** (lines 12-13) determines whether the model has converged or if training should continue for up to the specified number of rounds $T$. This step ensures that the iterative process continues until convergence criteria are met or the maximum number of rounds is reached.

**Algorithm 1** FL with three clients

```
1: Initialization:
2: Initialize global model parameters w
3: Initialize learning rate α, regularization strength λ, and number
   of training rounds T End Initialization
```

```
 4: for t = 1 to T do
 5:   for each client i do
 6:     Update local model parameters wᵢ based on the global model w and
   the local dataset (Xᵢ, Yᵢ)
 7:       wᵢ ← w−α∇L(w, Xᵢ, Yᵢ) + λ∇R(w)  ▷ Where L(w, Xᵢ, Yᵢ) is the local
   loss function, and R(w) is the regularization term
 8:     end for
 9:     After all clients have updated their local models, update the
   global model as follows:
10:       w ← (1/N)∑ᵢ₌₁ᴺ wᵢ
11: end for
12: Convergence Check:
13: Check for convergence or end the training after T rounds
14: End Convergence Check
```

## 3.6 Evaluation metrics

To test the results' reliability, the models were trained and tested using 10-fold cross-validation to identify the best-performing model. Finally, evaluation metrics were computed for each model to compare their performance. This study used $R^2$, RMSE, MAE, MAPE, and NSE as evaluation metrics to assess the machine learning model's performance. These metrics will help quantify how well the model estimates $ET_o$. $R^2$ measures the model's goodness of fit to the observed $ET_o$ values. It tells the proportion of the variance in $ET_o$ that the model can explain. The $R^2$ value can be obtained by Eq (5).

$$R^2 = 1 - \frac{SSR}{SST} \tag{5}$$

Where SSR presents the *sum of squared residuals* that can be computed by the squared difference between predicted and observed $ET_o$), and SST presents the total sum of squares that can be computed by estimating the squared difference between observed $ET_o$ and its mean).

RMSE presents the average magnitude of the errors between predicted and observed $ET_o$ values. RMSE of the proposed model prediction can be formalized as follows:

$$\text{RMSE} = \sqrt{\frac{1}{n}\sum_{i=1}^{n}\left(\text{Predicted } ET_0 - \text{Observed } ET_0\right)^2} \tag{6}$$

where predicted $ET_o$ presents the estimated $ET_o$ of the proposed model, Observed $ET_o$ presents the actual observed $ET_o$, and $n$ presents the number of data points. In this equation, $i$ is an index that corresponds to each specific data point in the dataset.

MAE can be utilized to measure the average absolute magnitude of the errors between predicted and observed $ET_o$ values (computed in the previous equation) can be formalized as follows:

$$\text{MAE} = \frac{1}{n}\sum_{i=1}^{n}\left|\text{Predicted } ET_0 - \text{Observed } ET_0\right| \tag{7}$$

where predicted $ET_o$ presents the estimated value of the proposed model, Observed $ET_o$ presents the estimated value measured from real-world data. Here, $i$ is an index that represents each data point in the dataset. The above equation aggregates all the individual errors into one total error, and $n$ represents the total number of data points in the given dataset.

MAPE can compute the average percentage difference between predicted and actual $ET_o$ values that can be formalized as follows:

$$MAPE = \frac{1}{n}\sum_{i=1}^{n}\frac{|\text{Predicted } ET_0 - \text{Observed } ET_0|}{|\text{Observed } ET_0|} \times 100 \tag{8}$$

Nash-Sutcliffe Efficiency (NSE), used to measure the performance of ML-based models. It assesses the predictive accuracy of a model by comparing the model's predictions to observed data. NSE mathematically expressed by (9)

$$NSE = 1 - \frac{\sum_{t=1}^{n}\left(\text{Predicted } ET_o - \text{Observed } ET_o\right)^2}{\sum_{t=1}^{n}\left(\text{Observed } ET_o - \overline{ET_o}\right)^2} \tag{9}$$

where $\overline{ET_o}$ is the mean of the observed $ET_o$ values, and $n$ is the total number of observations. Moreover, $t$ is an index representing each dataset observation.

## 4 Results

The study examined three machine learning models: RFR, DTR, and SVR. A proposed experiment uses three separate weather datasets from Multan, Faisalabad, and Rawalpindi. Performance was evaluated using key metrics, including $R^2$, RMSE, MAE, MAPE and NSE. A comparison of the performance of these models across different geographical locations is provided in the results. Notably, to our knowledge, the proposed FL method is the first to automatically estimate $ET_o$ for distributed fields using FL. Therefore, it is compared with traditional machine learning models instead of baseline models. At Multan, the RFR model achieved the highest $R^2$ and NSE value of 0.98 and MAPE of 6.72%. The obtained value of $R^2$ and MAPE indicates an excellent fit to the data. The lower RMSE values = 0.42, MAE = 0.32 mm day$^{-1}$ reveals the model's ability to accurately predict the $ET_o$. The obtained values of RMSE and MAE suggest precise and accurate $ET_o$ predictions with the RFR model. SVR and DTR model also performed good, with $R^2$ values above 0.95 and low in error metrics. A high $R^2$ and NSE value of 0.97 with the Faisalabad dataset using the RFR model is achieved.

For Faisalabad, the RFR outperformed other models with RMSE = 0.31, MAE = 0.23 mm day$^{-1}$, and MAPE of 5.21%. In the case of the Rawalpindi dataset, the RFR model outperformed other models with $R^2$ = 0.96, NSE = 0.96 and MAPE = 8.31%, indicating a good fit for the model. The RFR also exhibits a low RMSE = 0.37 and MAE = 0.28 mm day$^{-1}$ in $ET_o$ predictions. The DTR and SVR models also performed reasonably with $R^2$ values = 0.93, but they exhibit higher errors than the RFR model. Kruskal-Wallis test is also performed to evaluate the performance of RFR, SVR and DTR. The performance of three machine learning models is compared using evaluation metrics $R^2$, RMSE, MAE, MAPE, and NSE. The results of the test are described in Table 3. The p-values for all metrics are greater than 0.05. This indicates no significant difference in performance among the models across the local and the federated

**Table 3. Kruskal-Wallis test results.**

| Evaluation Metric | Test Statistic | P-Value |
|---|---|---|
| $R^2$ | 3.59 | 0.17 |
| RMSE | 5.68 | 0.06 |
| MAE | 4.93 | 0.08 |
| MAPE | 2.38 | 0.30 |
| NSE | 3.59 | 0.17 |

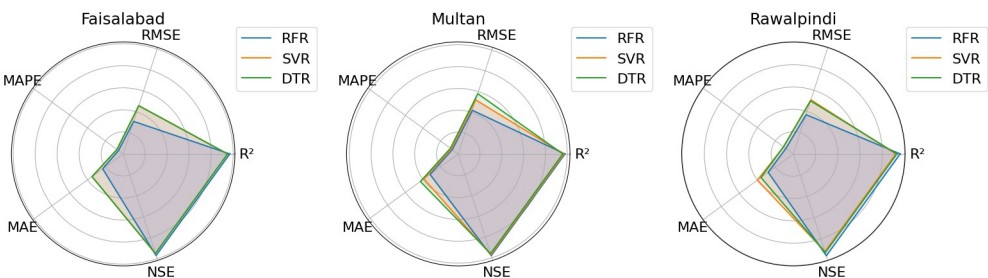

**Fig 15. Performance comparison of ML algorithms across multiple evaluation metrics.**

model. The RMSE and MAE values are closer to the threshold value. Moreover, we also perform the ANOVA for the reliability analysis. The results of ANOVA analysis suggest that f-ratio value is 14.6 and the p-value is.000077. The result is significant at $p < .05$.

Radar chart shows how the three models (RFR, SVR, DTR) perform according to the five metrics for each city as described in Fig 15. The radar charts represent that the models' performance across different metrics is relatively consistent within each city, with no significant outliers. In the federated approach described in Fig 16, RFR and SVR show better performance in terms of lower errors (MAE, MAPE, RMSE) and higher $R^2$ and NSE, while DTR has higher error metrics and lower $R^2$ and NSE values on average. Note that we selected a Radar chart instead of a Smith chart as a Radar chart provides a better understanding while comparing multiple variables across a single category.

The performance of different models in correlation and the standard deviation is shown in Taylor diagram 17, offering a comprehensive overview of model accuracy and variability.

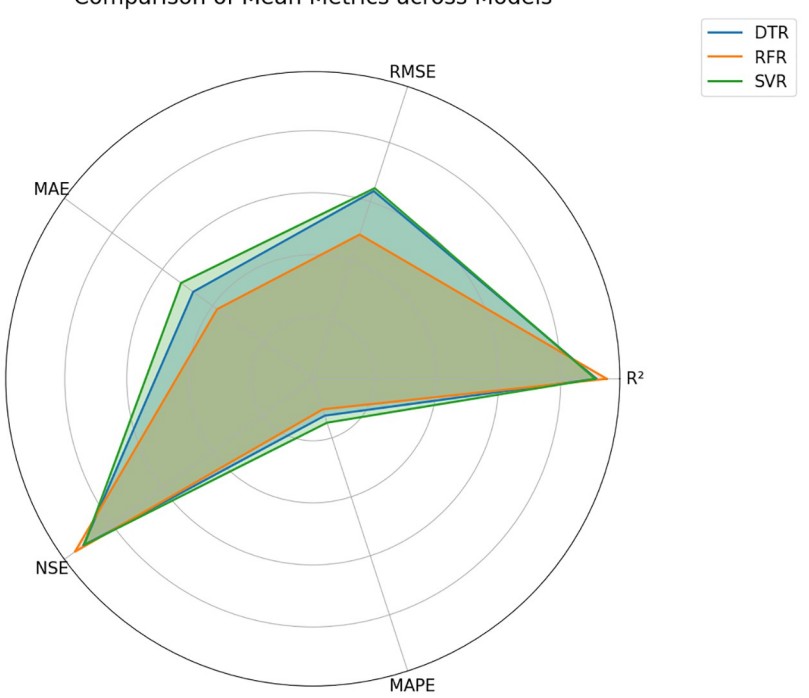

**Fig 16. Performance of federated learning on different ML models across multiple evaluation metrics.**

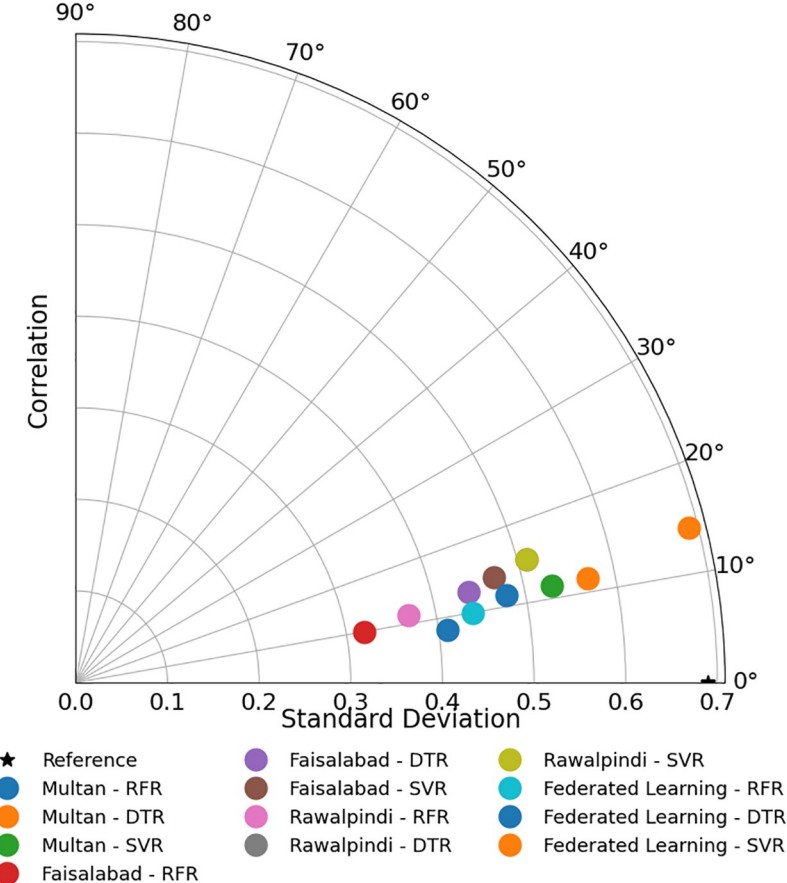

**Fig 17. Performance evaluation of different models using Taylor diagram on various datasets.**

Fig 17 suggests that the federated approach generally shows good performance with RFR and SVR, but DTR shows lower performance.

The error boxplot presented in Fig 18. It demonstrates that RFR constantly outperforms the DTR and SVR across all locations and performance metrics. RFR exhibits the highest $R^2$ and NSE values and the lowest MAPE, MAE, and RMSE values, representing greater predictive accuracy and reliability. DTR generally shows the poorest performance, with the lowest $R^2$ and NSE values and the highest MAPE, MAE, and RMSE values. SVR falls in between, performing better than DTR but not as well as RFR. FL models also show a similar trend, with RFR having the lowest MAPE values as represented in Fig 19. DTR in federated generally performs better than SVR in terms of RMSE and MAE, but worse in terms of $R^2$, NSE, and MAPE

A feature-importance-based analysis is also performed to determine the impact of different weather parameters on $ET_o$. The feature importance analysis is shown in Fig 20. By analyzing this information, we can better understand the relationship between weather features and $ET_o$. In feature analysis, it was found that Tmax and WS were the most influential parameters for $ET_o$ determination. A novel approach to learning called FL is also compared with separate traditional machine learning models in the study. This study also compared three regression models, RFR, SVR, and DTR, revealing varying performance metrics across evaluation criteria. The RFR outperformed other models with an $R^2$ value of 0.97 while maintaining lower errors with an RMSE of 0.44 and MAE of 0.33 mm day$^{-1}$ and MAPE of 8.18. The DTR results closely

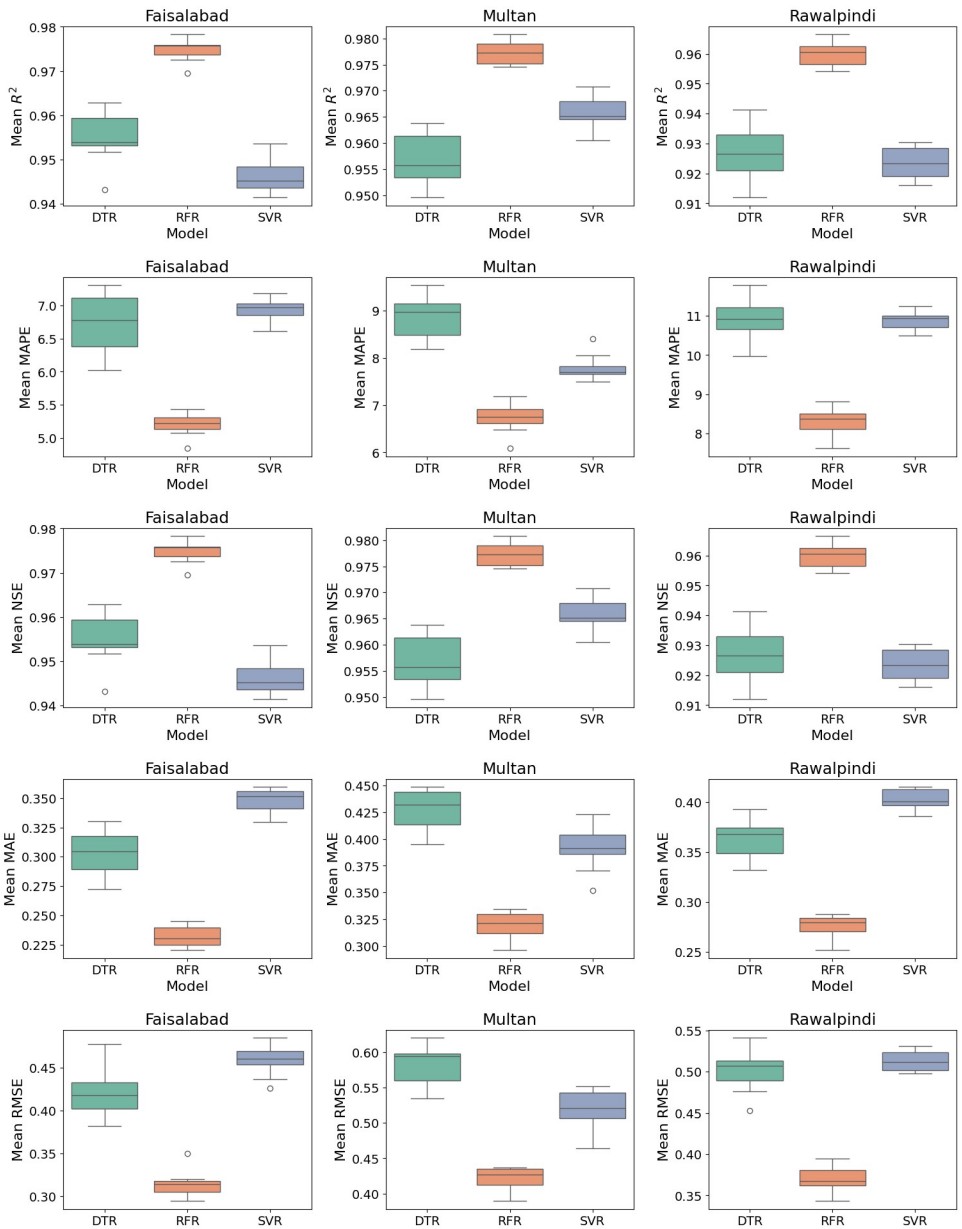

**Fig 18. Box plot analysis of evaluation metrics for RFR, DTR, and SVR models.**

followed the RFR results, with an $R^2$ = 0.96 and similar error values RMSE = 0.48, MAE = 0.35 mm day$^{-1}$ and MAPE = 8.50. This federated model, with an $R^2$ value = 0.97 and an RMSE = 0.44, can explain a large part of the changes in the target variable. In the FL approach, RMSE is higher than the best-performing individual model (e.g., Multan's RFR at RMSE 0.4064). Although the difference is relatively small, it still represents an accurate prediction. The MAE of the federated model = 0.33 mm day$^{-1}$, and the MAPE = 8.18%, demonstrating its ability to provide accurate estimates of ET$_o$. Performance metrics of different models across all the datasets are represented in Table 4.

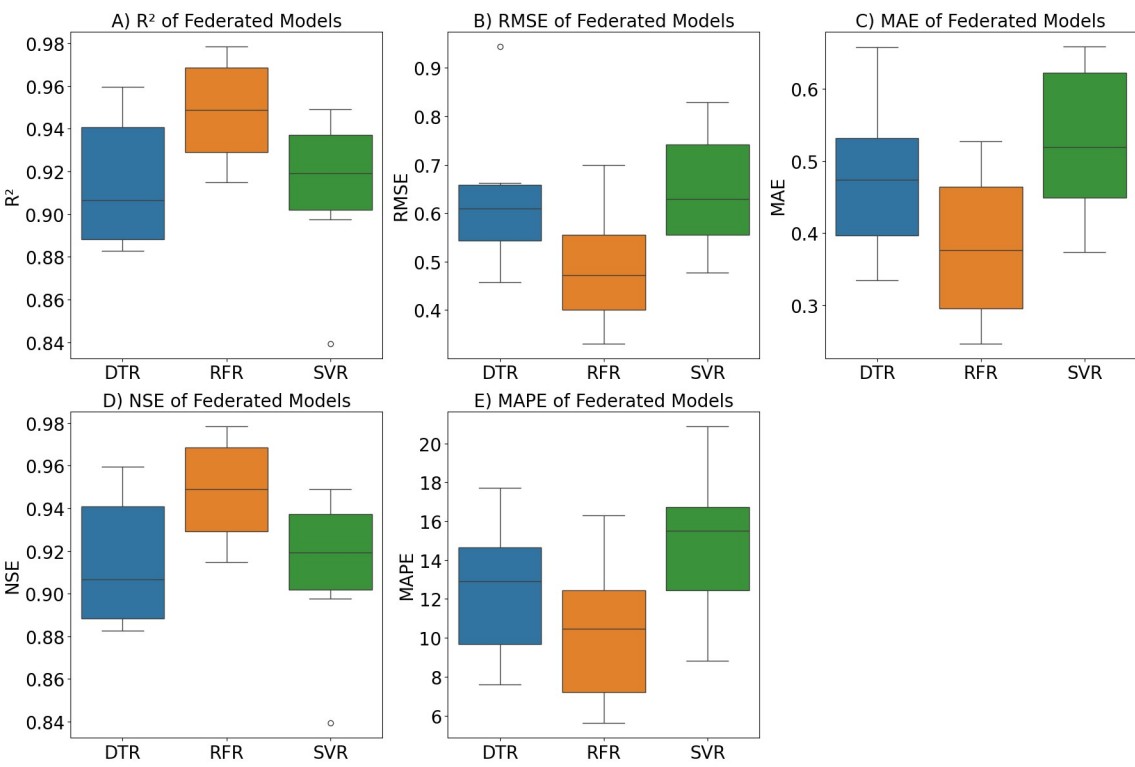

**Fig 19. Box plot analysis of evaluation metrics for RFR, DTR, and SVR models using federated learning.**

## 4.1 Discussion

The study implemented ML and FL models using Python and Google Colab by exploiting Python libraries: *Scikit-learn, Keras, and TensorFlow Federated.* The specific configurations included a processor with Single-core hyper-threaded Xeon Processors with RAM Memory of 12.72 GB, GPU of NVIDIA Tesla K80, P100, or T4 (depending on availability), providing

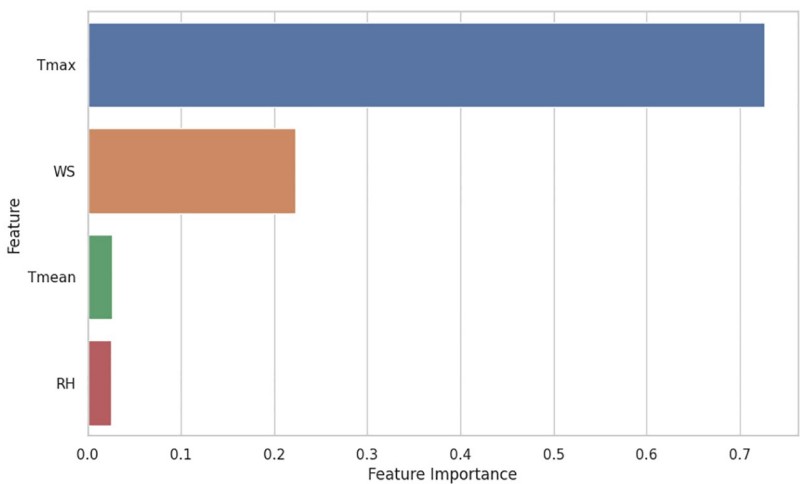

**Fig 20. Features importance analysis.**

**Table 4. Performance metrics of different models across datasets.**

| Model | Dataset | $R^2$ | RMSE | MAE | MAPE | NSE |
|---|---|---|---|---|---|---|
| RFR | Faisalabad | 0.97 | 0.31 | 0.23 | 5.21% | 0.97 |
| DTR | Faisalabad | 0.95 | 0.46 | 0.35 | 6.93% | 0.95 |
| SVR | Faisalabad | 0.95 | 0.46 | 0.35 | 6.93% | 0.95 |
| RFR | Multan | 0.98 | 0.42 | 0.32 | 6.72% | 0.98 |
| DTR | Multan | 0.96 | 0.58 | 0.43 | 8.86% | 0.96 |
| SVR | Multan | 0.97 | 0.52 | 0.39 | 7.79% | 0.97 |
| RFR | Rawalpindi | 0.96 | 0.37 | 0.28 | 8.31% | 0.96 |
| DTR | Rawalpindi | 0.93 | 0.50 | 0.36 | 10.95% | 0.93 |
| SVR | Rawalpindi | 0.92 | 0.51 | 0.40 | 10.88% | 0.92 |
| RFR | Federated | 0.95 | 0.49 | 0.38 | 10.35% | 0.95 |
| DTR | Federated | 0.91 | 0.64 | 0.48 | 12.49% | 0.91 |
| SVR | Federated | 0.91 | 0.65 | 0.53 | 14.88% | 0.91 |

substantial computational resources. While training time can vary due to internet connectivity and the availability of Google services. The RFR model consistently performed excellently across all three datasets, with high $R^2$ values and low error metrics. SVR and DTR models also produced competitive results, but often RFR outperformed them. Many parameters influence model selection, including the specific application, interpretability, and computational capacity. The FL approach combines data from multiple regions. It proved highly effective, especially when using the RFR. RFR is the most reliable model for accurate $ET_o$ estimation in diverse crop field settings. This finding is crucial for agricultural management, enabling more precise water resource optimization and planning. The high $R^2$ and low error metrics of RFR indicate its strong capability to handle the spatial variability in $ET_o$ data, making it a valuable tool for improving irrigation efficiency and crop yields. DTR showed more variability in its performance. Although DTR can capture non-linear relationships, it didn't perform as consistently as RFR. Regarding error metrics, the Support Vector Regressor (SVR) generally lagged behind RFR and DTR. It is likely due to its sensitivity to kernel choice and hyperparameters. The findings emphasize the importance of using the right machine-learning models for particular geographical locations when estimating $ET_o$ values. These findings explain how well machine learning models predict $ET_o$ for distributed crop fields. The RFR-based model seems appropriate for accurate $ET_o$ predictions, but the application's requirements should be considered when choosing the final model. This study also uses feature importance analysis to prioritize and select the most suitable features for the $ET_o$ prediction model. A machine learning model that emphasizes the most important factors may optimize training times and enhance interpretability by focusing on the most significant factors. Identifying traits of low value in data preparation and quality control might be useful. FL is beneficial when dealing with data distributed across multiple locations or clients, such as geographic regions. It allows models to be trained locally on specific datasets while preserving data privacy. Three different datasets (Multan, Faisalabad, and Rawalpindi) are combined to create the federated global model. The obtained results indicate strong performance across multiple evaluation metrics. The $R^2$ value of the federated model is comparable to that of the best-performing individual models on each dataset. It indicates the excellent generalization capabilities of the federated model. FL preserves privacy even though the RMSE of the federated model is slightly higher than that of the best individual models. The trade-off is acceptable when considering the accuracy of the predictions. As a result, the federated model's MAE and MAPE for estimating $ET_o$ across multiple

locations are reliable. Comparing FL to individual models for each dataset, the FL method can effectively predict $ET_o$. It can be beneficial when distributing data across multiple locations, and model generalization is a key concern. It is important to consider the particular requirements of the application when choosing between individual models and FL. Federated models may exhibit slightly higher RMSE, MAE, and MAPE, indicating a modest compromise in accuracy. However, this trade-off enhances their ability to generalize across diverse datasets, making them more robust, albeit less optimized for specific individual datasets. Distributed learning can indeed be an effective method for geographically dispersed data but in many real-world applications, transferring local weather data to a central location for model training may be infeasible due to bandwidth limitations, and data security regulations. Federated learning also addresses these concerns by allowing models to be trained locally at each site.

The study focuses on three locations in Pakistan, each characterized by distinct weather conditions. Expanding the research to encompass broader geographical areas could significantly enhance the model's adaptability and generalization. Moreover, Training a model with data from various regions, each with unique weather conditions, geographical features, and farming practices, is essential for achieving high accuracy. However, for future work, it is recommended to implement the proposed solution in regions with even more diverse weather conditions and to incorporate advanced deep learning approaches to refine the model's performance further.

## 5 Conclusion

The study proposed an FL approach for estimating reference evapotranspiration ($ET_o$) across multiple locations with distinct weather parameters. By employing various machine learning algorithms, including support vector machines, decision tree regression, and random forest regression (RFR), the research aimed to analyze and predict $ET_o$ effectively. The results demonstrated that the RFR model consistently outperformed other models at local and global levels, highlighting its robustness in $ET_o$ predictions. Feature importance analysis identified maximum temperature and wind speed as key weather parameters influencing $ET_o$ estimation. This research offers insights into the complex relationships between weather variables and $ET_o$. However, the model's adaptability might be limited by the study's focus on three specific locations in Pakistan. Future work should explore the application of this approach in regions with more diverse weather conditions and consider the integration of deep learning techniques for further improvement.

The study proposed an FL approach for $ET_o$ estimation of multiple locations with distinct weather parameters. Various machine learning algorithms were used to analyze and predict Reference Evapotranspiration ($ET_o$), including Support Vector Machines (SVM), DTR, and RFR. The implementation of the proposed solution reveals that the RER model outperformed local and global models with $R^2$ = 0.95, MAPE = 10.35, RMSE = 0.49, NSE = 0.95, and MAE = 0.38 (mm day$^{-1}$). The performance of the federate learning is satisfactory to estimate $ET_o$ with a single machine learning model trained using data of different locations. In the case of local models, the performance of the RFR model for the Multan dataset is $R^2$ = 0.98, MAPE = 6.72, RMSE = 0.42, NSE = 0.98 and MAE = 0.30 (mm day$^{-1}$). For the RFR model of the Faisalabad dataset is $R^2$ = 0.97, MAPE = 5.46, NSE = 0.97 RMSE = 0.32, and MAE = 0.24 (mm day$^{-1}$). For the RFR model of the Rawalpindi dataset is $R^2$ = 0.96, MAPE = 8.31, RMSE = 0.37, NSE = 0.96 and MAE = 0.27 (mm day$^{-1}$). Using a machine learning model, the RFR-based model outperformed the SVR and DTR in $ET_o$ predictions at global and local levels. A feature importance analysis revealed that maximum temperature and wind speed are the dominant weather parameters in $ET_o$ estimation. The study gains a deeper understanding of

the relationships between weather parameters and reference evapotranspiration. Three locations in Pakistan may limit the model's adaptability to other regions. Implementing the solution in areas with more diverse weather conditions and utilizing deep learning approaches are recommended for future work. The findings of this study underscore the potential of FL in enhancing ET predictions across varying climatic conditions, paving the way for improved agricultural management practices.

## Author Contributions

**Conceptualization:** Muhammad Tausif, Muhammad Waseem Iqbal.

**Data curation:** Muhammad Tausif.

**Formal analysis:** Muhammad Tausif, Amjad Rehman Khan.

**Funding acquisition:** Alex Elyassih, Amjad Rehman Khan.

**Investigation:** Alex Elyassih.

**Methodology:** Muhammad Tausif.

**Project administration:** Muhammad Tausif, Muhammad Waseem Iqbal.

**Resources:** Muhammad Tausif.

**Software:** Muhammad Tausif, Bayan AlGhofaily.

**Supervision:** Muhammad Waseem Iqbal, Rab Nawaz Bashir.

**Validation:** Rab Nawaz Bashir.

**Visualization:** Bayan AlGhofaily.

**Writing – review & editing:** Muhammad Tausif.

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
