## [Decision Letter · Decision Letter 0]

24 May 2024

PONE-D-24-15636Federated learning based reference evapotranspiration estimation for distributed crop fieldsPLOS ONE

Dear Dr. Tausif,

Thank you for submitting your manuscript to PLOS ONE. After careful consideration, we feel that it has merit but does not fully meet PLOS ONE’s publication criteria as it currently stands. Therefore, we invite you to submit a revised version of the manuscript that addresses the points raised during the review process.

We look forward to receiving your revised manuscript.

Kind regards,

Nguyen-Thanh Son, Ph.D.

Academic Editor

PLOS ONE

Journal Requirements:

"The research is supported by AIDA Lab CCIS Prince sultan University Riyadh Saudi Arabia

The author would like to thank Prince Sultan University, Riyadh Saudi Arabia for support of APC for this publication."

4. Please expand the acronym “AIDA” (as indicated in your financial disclosure) so that it states the name of your funders in full.

"The research is supported by AIDA Lab CCIS Prince sultan University Riyadh Saudi Arabia

The author would like to thank Prince Sultan University, Riyadh Saudi Arabia for support of APC for this publication."

"The research is supported by AIDA Lab CCIS Prince sultan University Riyadh Saudi Arabia

The author would like to thank Prince Sultan University, Riyadh Saudi Arabia for support of APC for this publication."

6. Thank you for stating the following in your Competing Interests section:  

"The authors declare that they have no known competing financial interests or personal relationships that could have appeared to influence the work reported in this paper."

7. Please provide a complete Data Availability Statement in the submission form, ensuring you include all necessary access information or a reason for why you are unable to make your data freely accessible. If your research concerns only data provided within your submission, please write "All data are in the manuscript and/or supporting information files" as your Data Availability Statement.

8. Please amend either the abstract on the online submission form (via Edit Submission) or the abstract in the manuscript so that they are identical.

9. We note that [Figure 2] in your submission contain [map/satellite] images which may be copyrighted. All PLOS content is published under the Creative Commons Attribution License (CC BY 4.0), which means that the manuscript, images, and Supporting Information files will be freely available online, and any third party is permitted to access, download, copy, distribute, and use these materials in any way, even commercially, with proper attribution. For these reasons, we cannot publish previously copyrighted maps or satellite images created using proprietary data, such as Google software (Google Maps, Street View, and Earth). For more information, see our copyright guidelines: http://journals.plos.org/plosone/s/licenses-and-copyright.

a. You may seek permission from the original copyright holder of Figure [2] to publish the content specifically under the CC BY 4.0 license.  

Reviewers' comments:

Reviewer's Responses to Questions

**Comments to the Author**

1. Is the manuscript technically sound, and do the data support the conclusions?

Reviewer #1: Partly

Reviewer #2: Yes

2. Has the statistical analysis been performed appropriately and rigorously? 

Reviewer #1: Yes

Reviewer #2: I Don't Know

3. Have the authors made all data underlying the findings in their manuscript fully available?

Reviewer #1: Yes

Reviewer #2: Yes

4. Is the manuscript presented in an intelligible fashion and written in standard English?

Reviewer #1: Yes

Reviewer #2: Yes

5. Review Comments to the Author

Reviewer #1: ---The purpose section should be emphasized better.

---Give information about the agriculture of the study area.

---Write the limitations of the methods.

---Add the limitations of the study and your suggestions to the results.

---It needs a good discussion section separate from results.

---Test the reliability of prediction results.

--- Compare the measurement and prediction results by performing the Kruskal-Wallis test.

--- Draw the Taylor diagram.

--- Draw error boxplot and violin graphs.

---Add the scatter diagrams.

---Add descriptive statistics parameters.

--- Flowchart is quite pale, needs some improvement.

--- Which software did you analyze with?

--- Can you give information about analysis times? Also provide information about the computer on which you performed the analysis.

--- Include descriptive statistics of the data.

--- Add NSE to the Evaluation criteria section. Interpret the MAPE, R2 and NSE criteria according:

Uncuoglu, E., Citakoglu, H., Latifoglu, L., Bayram, S., Laman, M., Ilkentapar, M., & Oner, A. A. (2022). Comparison of neural network, Gaussian regression, support vector machine, long short-term memory, multi-gene genetic programming, and M5 Trees methods for solving civil engineering problems. Applied Soft Computing, 129, 109623.

--- Check out the following articles about

Bayram, S., & Çıtakoğlu, H. (2023). Modeling monthly reference evapotranspiration process in Turkey: application of machine learning methods. Environmental Monitoring and Assessment, 195(1), 67.

Citakoglu, H., Cobaner, M., Haktanir, T., & Kisi, O. (2014). Estimation of monthly mean reference evapotranspiration in Turkey. Water Resources Management, 28, 99-113.

Cobaner, M., Citakoğlu, H., Haktanir, T., & Kisi, O. (2017). Modifying Hargreaves–Samani equation with meteorological variables for estimation of reference evapotranspiration in Turkey. Hydrology Research, 48(2), 480-497.

Reviewer #2: In the present paper, the authors propose an interesting approach for reference evapotranspiration estimation for distributed crop fields using Federated learning based techniques. The paper seem to be interesting and needs to be improved before publication. Following are the major and minor and improvements:

Major comments

I. The introduction lacks recent literature updates. It is essential to incorporate recent studies to provide a comprehensive overview of the current state of research in the field. Below are suggested paragraphs to be added to the literature review section to enhance the introduction:

"Machine learning Remote sensing field becomes cost effective tools in modeling hydrological variables e.g. River discharge, rainfall runoff estimation, Evapotranspiration, land use classification, surface temperature estimation". Following studies will enrich the introduction and literature sections:

1. https://doi.org/10.3390/rs12071107

2. https://doi.org/10.1029/2023WR035785

3. https://link.springer.com/article/10.1007/s43621-024-00263-w

4. https://doi.org/10.1201/9781003377825 (Chapter 3)

5. https://doi.org/10.1007/s42452-023-05520-7

6. https://doi.org/10.1186/s40562-023-00287-6

7. https://doi.org/10.3390/su15043572

8. https://doi.org/10.1007/978-3-031-29394-8_8

9. https://doi.org/10.3390/atmos13101609

II. Novelty of the paper is not written clearly. It should be highlighted.

III. How you divide the dataset? How much data is used for training and testing periods? I suggest authors to add table along with statistics in dataset sections.

IV. I did not find adequate information on the methodologies related to Random Forest Regressor (RFR), Support Vector Regressor (SVR), and Decision Tree Regressor (DTR). Although I found some information on the Federated Learning (FL) model, it is not sufficient for the focus of this paper, which centers around Federated Learning.

V. Compare your findings with previous stud and visual representation (Taylor diagram, Radar chart, Smith graph) of machine learning results should be added. Below are the reference studies and need to be citied:

1. https://doi.org/10.1007/s13201-022-01667-7

2. https://www.mdpi.com/2073-4441/14/10/1666

3. https://www.mdpi.com/2306-5338/10/8/169

4. https://www.mdpi.com/2073-4441/15/6/1149

5. https://doi.org/10.1002/ird.2838

Minor comments

Some grammatical errors are present and need revision.

Clear location map in Figure 2 is needed.

Improvement in the graphical representation of Figure 3.

Check the references as per journal guidelines and add DOI in references

6. PLOS authors have the option to publish the peer review history of their article (what does this mean?). If published, this will include your full peer review and any attached files.

Reviewer #1: No

Reviewer #2: No

---

## [Author Response · Author response to Decision Letter 0]

13 Jun 2024

Thank you for the comments from the reviewers and editor on how to improve our manuscript. The response to reviewer and editor comments are attached in the response-to-reviewer file attached in the submission.

---

## [Decision Letter · Decision Letter 1]

2 Aug 2024

PONE-D-24-15636R1Federated learning based reference evapotranspiration estimation for distributed crop fieldsPLOS ONE

Dear Dr. Tausif,

Thank you for submitting your manuscript to PLOS ONE. After careful consideration, we feel that it has merit but does not fully meet PLOS ONE’s publication criteria as it currently stands. Therefore, we invite you to submit a revised version of the manuscript that addresses the points raised during the review process.

One or more of the reviewers has recommended that you cite specific previously published works. Members of the editorial team have determined that the works referenced are not directly related to the submitted manuscript. As such, please note that it is not necessary or expected to cite the works requested by the reviewer.

We look forward to receiving your revised manuscript.

Kind regards,

Hanna Landenmark

Staff Editor

PLOS ONE

on behalf of 

Nguyen-Thanh Son

Journal Requirements:

Additional Editor Comments:

One or more of the reviewers has recommended that you cite specific previously published works. Members of the editorial team have determined that the works referenced are not directly related to the submitted manuscript. As such, please note that it is not necessary or expected to cite the works requested by the reviewer.

Reviewers' comments:

Reviewer's Responses to Questions

**Comments to the Author**

1. If the authors have adequately addressed your comments raised in a previous round of review and you feel that this manuscript is now acceptable for publication, you may indicate that here to bypass the “Comments to the Author” section, enter your conflict of interest statement in the “Confidential to Editor” section, and submit your "Accept" recommendation.

Reviewer #1: All comments have been addressed

Reviewer #3: (No Response)

Reviewer #4: (No Response)

2. Is the manuscript technically sound, and do the data support the conclusions?

Reviewer #1: Yes

Reviewer #3: (No Response)

Reviewer #4: Partly

3. Has the statistical analysis been performed appropriately and rigorously? 

Reviewer #1: Yes

Reviewer #3: (No Response)

Reviewer #4: No

4. Have the authors made all data underlying the findings in their manuscript fully available?

Reviewer #1: Yes

Reviewer #3: (No Response)

Reviewer #4: No

5. Is the manuscript presented in an intelligible fashion and written in standard English?

Reviewer #1: Yes

Reviewer #3: (No Response)

Reviewer #4: Yes

6. Review Comments to the Author

Reviewer #1: The authors of the article have satisfactorily incorporated all my comments and I have no further comments.

Reviewer #3: 1. Time spent need to be measured in the experimental results.

2. Limitation and Discussion Sections need to be inserted.

3. All metrics need to be calculated in the experimental results.

4. The parameters used for the analysis must be provided in table

5. Comparison with similar studies on a similar dataset need to be inserted (with references).

6. The cost associated with deploying these deep learning models, including the necessary hardware and software, is not addressed.

7. The architecture of the proposed model must be provided

8. The authors need to make a clear proofread to avoid grammatical mistakes and typo errors.

9. Enhance the clarity of the Figures by improving their resolution.

10. Add future work in last section (conclusion) (if any)

11. To improve the Related Work and Introduction sections authors are recommended to review this highly related research work paper:

a) Feature reduction for hepatocellular carcinoma prediction using machine learning algorithms

b) Detecting cyberbullying using deep learning techniques: a pre-trained glove and focal loss technique

c) Optimal Gasoline Price Predictions: Leveraging the ANFIS Regression Model

d) The effect of rebalancing techniques on the classification performance in cyberbullying datasets

e) Developing an efficient method for automatic threshold detection based on hybrid feature selection approach

f) Acoustic data detection in large-scale emergency vehicle sirens and road noise dataset

g) Employing machine learning for enhanced abdominal fat prediction in cavitation post-treatment

h) Predicting female pelvic tilt and lumbar angle using machine learning in case of urinary incontinence and sexual dysfunction

Reviewer #4: The manuscript used estimated reference evapotranspiration of multiple locations with distinct weather conditions using a federated learning approach, which allows for training ML models across multiple decentralized devices or servers holding local data samples, without exchanging them. They applied three ML models, including Random Forest Regressor (RFR), Support Vector Regressor (SVR), and Decision Tree Regressor (DTR), for each location. The review comments are presented in the following:

Title:

1. I do not understand why you recommend federated learning over centralized learning for estimating reference evapotranspiration. The title is “Federated learning based reference evapotranspiration estimation for distributed crop fields”. It means that you are proposing a federated learning, which is suitable for heterogeneous datasets whose sizes may span several orders of magnitude, for estimating a variable in three location (distributed) with quite similar datasets. What is wrong with applying distributed learning for estimating reference evapotranspiration?

Abstract:

1. The abstract does not justify why there is a need to apply a federated learning for estimating reference evapotranspiration.

Introduction:

2. The gap in the literature is not established in the introduction.

3. Remove Equation 1 from the introduction and present it in the materials and methods section.

4. It is stated in lines 35-37 that “Moreover, simple ML approaches also face limitations, particularly related to data centralization and may not effectively capture the diverse climatic conditions of different regions.” For this case, why do not we add climatic conditions as an input instead of considering a federated learning for estimating reference evapotranspiration.

5. I suggest merging the section of“2 Related work” within the introduction.

6. Lines 177-184: This paragraph is repeating the last paragraph of the introduction. Avoid repeating.

Methods:

7. Lines 285-286: “…. which has proven to be a viable solution for overcoming the challenges mentioned earlier and improving ETo estimates.” This sentence requires a reference. Otherwise, you have to justify it.

8. There are a lot figures in this section, while very little description is provided in the text. Explain what every figure/table shows in details in the text.

9. Figure 14 is not a good flowchart for the FL. It does not show how you integrate the local models.

10. Line 301: Remove ” from the end of the sentence.

11. Check Equation 3 and introduce all parameters in this equation appropriately.

12. Lines 303-304: Where is the description of this part?

13. Add an appropriate reference for any equation that it is not from your work.

14. More information should be added regarding how you calibrate the hyperparameters of the ML models.

15. Which software/libraries you used to implement the ML algorithms and FL?

16. Add a reliability analysis (10.1007/s40899-023-01021-y) to check how reliable the estimations made by your ML models are.

Results:

17. How much did FL model improve estimated reference evapotranspiration in comparison to using individual ML models?

18. There are a lot figures in this section, while very little description is provided in the text. Explain what every figure/table shows in details in the text.

19. What are the advantages and disadvantages of the FL model?

Discussion

20. What is the practical contribution of your work? How does it contribute to the practical field in the realm of improving water balance analysis?

21. The discussion should merge the findings of this work with other ones available in the literature.

Conclusion:

22. The conclusion should contain general findings from the study and present some take-home messages. Improve this section.

7. PLOS authors have the option to publish the peer review history of their article (what does this mean?). If published, this will include your full peer review and any attached files.

Reviewer #1: No

Reviewer #3: No

Reviewer #4: No

---

## [Author Response · Author response to Decision Letter 1]

20 Aug 2024

Response to comments and suggestions by reviewers: We sincerely thank the reviewers for providing detailed comments, which have helped us to improve our manuscript’s presentation and quality. In the following, we provide a detailed response and changes made for each comment.

As the paper has been revised, it is difficult to append verbatim changes for each response. We have categorically specified the places in the paper which have been changed at the end of each response.

Response to Editor’s Comments:

R2.1: Abbreviations in Table 1: Many abbreviations are used and not included in the Abbreviations List.

Response: thank you for your comment. Accordingly, the missing abbreviations are added in the Section ”Abbreviations and Acronyms” (Page 15, Lines 523-542).

R2.2: Line 211: Specify which observed data were collected from the selected stations and which equation was used to calculate the daily reference ETo.

Response: We apologize for neglecting the issue. Accordingly, we have explicitly specified the used equation in Section 3.2 (Page 6, Line2 213-214).

R2.3: Equations 5 to 8: The index ”I” or ”t” is missing.

Response: Thank you for your valuable comment. We have revised the manuscript accordingly (Section 3.5, Pages 10-11, Lines 346-47, 352-53, 362).

R2.4: Figure 17: The Taylor diagram is incorrectly presented; the x and y axes should be the standard deviation (STD), and the radial axis should represent the correlation (R). Response: Thank you for your comment. Our Taylor diagram accurately follows standard conventions, with the radial axis representing standard deviation (STD) and the angular coordinate showing correlation (R). This structure is consistent with established practices, as detailed in Kenneth E. Taylor’s ”Summarizing Multiple Aspects of Model Performance in a Single Diagram” (2001). If specific aspects of our diagram seem unclear, we are open to further clarification or adjustments as needed.

R2.5: Figure 21: This figure does not provide additional information. The results in Table 4 are more informative.

Response: Thank you for your valuable comment. Accordingly, we removed the manuscript’s Figure 21.

R2.6: Sentences (from line 54 to 57 and from 284 to 286 ) need references

Response: The references [17, 63] are added. Thank you for your comment.

Response to Reviewer 3 Comments:

R3.1: Time spent need to be measured in the experimental results.

Response: Thank you for your comment. Accordingly, we have explicitly mentioned the time spent in Section 4.1 (Page 12, Lines 432-437).

R3.2: Limitation and Discussion Sections need to be inserted.

Response: Thank you for your valuable suggestion. Accordingly, we have added the Limitations in the Last Paragraph of Section 4.1 (Page 14, Lines 480-488).

R3.3: All metrics need to be calculated in the experimental results.

Response: Thank you for your comment. All the metrics are calculated and reported in the experimental results (Table 4 (Page 13)) and discussed in the Last Paragraph of Section 4 (Page 12, Lines 418-430).

R3.4: The parameters used for the analysis must be provided in table.

Response: Thank you for your comment. The parameters used for analysis and their statistic analysis are provided in Table 2 (Page 7).

R3.5: Comparison with similar studies on a similar dataset need to be inserted (with references).

Response: Thank you for your comment. Notably, to our knowledge, the proposed FL method is the first to automatically estimate ETo for distributed fields using Federated Learning (FL). Therefore, it is compared with traditional machine learning models instead of baseline models.

We have explicitly specified it in Section 4 (Pages 11, Lines 368-371).

R3.6:The cost associated with deploying these deep learning models, including the necessary hardware and software, is not addressed.

Response: Thank you for your comment. Accordingly, we have addressed the cost of deploying these deep learning models, including the necessary hardware and software in Section

4.1 (Page 12, Line 432-437).

R3.7: The architecture of the proposed model must be provided.

Response: Thank you for your valuable suggestion. Although the architecture is illustrated in Figure 1 and Figure 14, and its working is presented in Algorithm 1, we have explicitly added and explained the architecture of the proposed model in the explanation of Algorithm 1 (Page 9-10, Lines 316-329).

R3.8: The authors need to make a clear proofread to avoid grammatical mistakes and typo errors.

Response: Thank you for your comment. Accordingly, we have proofread to avoid grammatical and typo mistakes.

R3.9: Enhance the clarity of the Figures by improving their resolution.

Response: Fixed. Thank you for your comment.

R3.10: Add future work in last section (conclusion) (if any).

Response: Thank you for your comment. The manuscript is revised accordingly (Section 5, Page 14, Lines 500-502).

R3.11: To improve the Related Work and Introduction sections authors are recommended to review this highly related research work paper.

a) Feature reduction for hepatocellular carcinoma prediction using machine learning algo-rithms

b) Detecting cyberbullying using deep learning techniques: a pre-trained glove and focal losstechnique

c) Optimal Gasoline Price Predictions: Leveraging the ANFIS Regression Model

d) The effect of rebalancing techniques on the classification performance in cyberbullyingdatasets

e) Developing an efficient method for automatic threshold detection based on hybrid featureselection approach

f) Acoustic data detection in large-scale emergency vehicle sirens and road noise dataset

g) Employing machine learning for enhanced abdominal fat prediction in cavitation post-treatment

h) Predicting female pelvic tilt and lumbar angle using machine learning in case of urinaryincontinence and sexual dysfunction

Response: Thank you for your suggestions to improve the work. Due to editor instruction, we are unable to incorporate changes. Editor has restricted in this manner. Hope you will understand.

Response to Reviewer 4 Comments:

R4.1: I do not understand why you recommend federated learning over centralized learning for estimating reference evapotranspiration. The title is “Federated learning based reference evapotranspiration estimation for distributed crop fields”. It means that you are proposing a federated learning, which is suitable for heterogeneous datasets whose sizes may span several orders of magnitude, for estimating a variable in three location (distributed) with quite similar datasets. What is wrong with applying distributed learning for estimating reference evapotranspiration.

Response: Thank your for your comment. By training local models and sharing their insights, federated learning helps create more accurate ETo estimates that work well in different locations. Even if the datasets look similar, small differences in local conditions, soil types, or weather patterns can impact ETo estimates. The detailed reasons for choosing federated learning over machine/deep learning are mentioned in Section 1 (Page 2, Lines 49-55).

R4.1: The abstract does not justify why there is a need to apply a federated learning for estimating reference evapotranspiration.

Response: Thank you for your comment. We have revised the Abstract accordingly.

R4.2: The gap in the literature is not established in the introduction.

Response: Thank you for your comment. We have established the gap in the literature in Introduction (Section 1, Page 2, Lines 35-49). .

R4.3: Remove Equation 1 from the introduction and present it in the materials and methods section.

Response: Equation 1 is moved from the Introduction (Section 1) to materials and methods (Section 3, Equation 1, Lines 184-185).

R4.4: It is stated in lines 35-37 that “Moreover, simple ML approaches also face limitations, particularly related to data centralization and may not effectively capture the diverse climatic conditions of different regions.” For this case, why do not we add climatic conditions as an input instead of considering a federated learning for estimating reference evapotranspiration.

Response: We apologize for neglecting the issue. Accordingly, we have explicitly specified the reason why we do not add climatic conditions as an input instead of considering federated learning for estimating reference evapotranspiration in Section 3.2 (Page 7, Lines 220222).

R4.5: I suggest merging the section of “2 Related work” within the introduction.

Response: Thank you for your suggestion. Notable, we have separated the Introduction and Related Work sections as per the requirements of the PLoS One journal.

R4.6: 177-184: This paragraph is repeating the last paragraph of the introduction. Avoid repeating. Methods:

Response: Thank you for your valuable suggestion. The reputation is removed as suggested (Section 2, Page 5, Lines 177-179).

R4.7: Lines 285-286: “.... which has proven to be a viable solution for overcoming the challenges mentioned earlier and improving ETo estimates.” This sentence requires a reference. Otherwise, you have to justify it.

Response: The reference [63] is added. Thank you for your comment.

R4.8: There are a lot figures in this section, while very little description is provided in the text. Explain what every figure/table shows in details in the text.

Response: Thank you for your valuable suggestion. We have revised the manuscript accordingly (Section 3.2, Page 6-8, Lines (217-219, 234-238, 247-248, 258-262)).

R4.9: Figure 14 is not a good flowchart for the FL. It does not show how you integrate the local models.

Response: Thank you for your comment. Accordingly, we have revised Figure 14.

R4.10: Line 301: Remove ” from the end of the sentence.

Response: Fixed. Thank you for your comment.

R4.11: Check Equation 3 and introduce all parameters in this equation appropriately.

Response: Thank you for your comment. Accordingly, the parameters of equation 3 are defined in Section 3.4 (Page 9, Lines 313-315).

R4.12: Lines 303-304: Where is the description of this part?.

Response: Thank you for your valuable comment. Accordingly, we have added the description of Algorithm 1 (Pages 9-10, Lines 316-329).

R4.13: Add an appropriate reference for any equation that it is not from your work

Response: Thank you for your comment. Accordingly, we have double-checked the equations used from the literature and cited them wherever required.

R4.14: More information should be added regarding how you calibrate the hyperparameters of the ML models.

Response: Thank you for your comment. Accordingly, the suggested information is added in Section 3.4 (Pages 9-10, Lines 316-319).

R4.15: Which software/libraries you used to implement the ML algorithms and FL?.

Response: Thank you for your comment. We have explicitly specified the libraries used to implement ML/FL algorithms in Section 4.1 (Page 12, Lines 432-433),

R4.16: Add a reliability analysis (10.1007/s40899-023-01021-y) to check how reliable the estimations made by your ML models are.

Response: Thank you for your comment. The analysis presented, while thorough and insightful, is not well-suited for implementation within a federated learning framework. As a result, this approach was not included in the study. However, we acknowledge the value of exploring alternative methods and remain open to considering other techniques in future research.

R4.17: How much did FL model improve estimated reference evapotranspiration in comparison to using individual ML models?.

Response: Thank you for your comment. Accordingly, we have revised the manuscript (Section 4.1, Pages 13-14, Lines 476-479).

R4.18: There are a lot figures in this section, while very little description is provided in the text. Explain what every figure/table shows in details in the text

Response: Thank you for your valuable suggestion. We have revised the manuscript accordingly (Section 3.2, Page 6-8, Lines (217-219, 234-238, 247-248, 258-262)).

R4.19: What are the advantages and disadvantages of the FL model?

Response: Thank you for your comment. The manuscript is revised accordingly (Section 1, Page 2-3, 56-64).

R4.20: What is the practical contribution of your work? How does it contribute to the practical field in the realm of improving water balance analysis.

Response: Thank you for your comment. The manuscript is revised accordingly (Section 1, Page 3, 64-67).

R4.21: The discussion should merge the findings of this work with other ones available in the literature.

Response: Thank you for your comment. Notably, to our knowledge, the proposed FL method is the first to automatically estimate ETo for distributed fields using Federated Learning (FL). Therefore, it is compared with traditional machine learning models instead of models available in the literature.

R4.22: The conclusion should contain general findings from the study and present some take-home messages. Improve this section.

Response: Thank you for your valuable suggestion. The conclusion section (Page 14, Lines 490-502) is revised accordingly.

---

## [Decision Letter · Decision Letter 2]

16 Sep 2024

PONE-D-24-15636R2Federated learning based reference evapotranspiration estimation for distributed crop fieldsPLOS ONE

Dear Dr. Tausif,

Thank you for submitting your manuscript to PLOS ONE. The reviewers recommend reconsideration of your manuscript following major revision. After careful consideration, we feel that it has merit but does not fully meet PLOS ONE’s publication criteria as it currently stands. Therefore, we invite you to submit a revised version of the manuscript that addresses the points raised during the review process. 

We look forward to receiving your revised manuscript.

Kind regards,

Ghani Rahman

Academic Editor

PLOS ONE

**Additional Editor Comments:**

I have completed my evaluation of your manuscript. The reviewers recommend reconsideration of your manuscript following major revision. I invite you to resubmit your manuscript after addressing the reviewer comments.

Reviewers' comments:

Reviewer's Responses to Questions

**Comments to the Author**

1. If the authors have adequately addressed your comments raised in a previous round of review and you feel that this manuscript is now acceptable for publication, you may indicate that here to bypass the “Comments to the Author” section, enter your conflict of interest statement in the “Confidential to Editor” section, and submit your "Accept" recommendation.

Reviewer #1: All comments have been addressed

Reviewer #4: (No Response)

Reviewer #5: (No Response)

2. Is the manuscript technically sound, and do the data support the conclusions?

Reviewer #1: Yes

Reviewer #4: No

Reviewer #5: No

3. Has the statistical analysis been performed appropriately and rigorously? 

Reviewer #1: Yes

Reviewer #4: No

Reviewer #5: N/A

4. Have the authors made all data underlying the findings in their manuscript fully available?

Reviewer #1: Yes

Reviewer #4: No

Reviewer #5: Yes

5. Is the manuscript presented in an intelligible fashion and written in standard English?

Reviewer #1: Yes

Reviewer #4: No

Reviewer #5: Yes

6. Review Comments to the Author

**Reviewer #1:** The authors of the article have satisfactorily incorporated all my comments and I have no further comments.

**Reviewer #4: **The authors' responses are not detailed and some of the major comments were not addressed properly. For instance, they state in lines 41-42: "A universal ETo model cannot be created using traditional machine learning methods because they can’t handle diverse weather conditions."This is not correct. There are many papers on application of traditional machine learning methods for this purpose. It is not the gap in the literature. In addition, the authors' response to the comment why they applied "federated learning over centralized learning for estimating reference evapotranspiration" is not acceptable. It is important to note that there are many machine learning models and artificial intelligence techniques. Nevertheless, they should be used for solving appropriate problems. In other words, the authors proposed federated learning for estimating reference evapotranspiration, while it is not justified. Instead of federated learning, distributed learning, which is a suitable tool. can be used. Since the major contribution of this manuscript is proposinjg federated learning for estimating reference evapotranspiration, I suggest rejection.

**Reviewer #5:** 1. The author mentions that a "feature importance-based analysis" was conducted; however, the manuscript lacks details on how the importance of the features was calculated. Please provide a clear explanation and justification for the method used to determine feature importance, including any metrics, algorithms, or criteria employed in the analysis.

2. The manuscript lacks technical details about the proposed Federated Learning-based framework for estimating ETo. Readers would benefit from more specific information about the framework's design and implementation. For example, what type of architectural design was used on the client side in FL environment for estimating ETo? How many clients participated in the training process? How many communication rounds occurred between the server and clients to exchange trained weights? Additionally, if machine learning algorithms such as Random Forest Regressor (RFR), Decision Tree Regressor (DTR), or Support Vector Regressor (SVR) were trained on the client side, it is important to explain how the trained weights were aggregated on the server side, evaluated, and then sent back to the clients. Please provide appropriate details to address these concerns, and if possible, include code to further clarify your approach.

7. PLOS authors have the option to publish the peer review history of their article (what does this mean?). If published, this will include your full peer review and any attached files.

Reviewer #1: No

Reviewer #4: No

Reviewer #5: No

---

## [Author Response · Author response to Decision Letter 2]

3 Oct 2024

Dear Reviewers and Editor

I hope this message finds you well. We are pleased to submit the revised version of our manuscript. Thank you for your insightful feedback on our manuscript. We appreciate the time and effort you dedicated to your reviews. We sincerely appreciate the valuable feedback provided by you and the reviewers. We have carefully addressed all comments and suggestions, which we believe have significantly improved the manuscript. Below, we outline the key changes made in response to the reviewers' feedback:

Reviewer #1:

We are grateful for your positive comments regarding “all comments are addressed”and for recognizing our efforts in addressing previous concerns. Your support is invaluable.

Reviewer #4:

Thank you for your thorough review. We appreciate your points on federated learning (FL). We have clarified our justification for FL over traditional methods, emphasizing its advantages in privacy preservation and collaborative training without sharing raw data. We have expanded this discussion in the revised manuscript.

Comment 1: 

The authors' responses are not detailed and some of the major comments were not addressed properly. For instance, they state in lines 41-42: "A universal ETo model cannot be created using traditional machine learning methods because they can’t handle diverse weather conditions."This is not correct. There are many papers on application of traditional machine learning methods for this purpose. It is not the gap in the literature. In addition, the authors' response to the comment why they applied "federated learning over centralized learning for estimating reference evapotranspiration" is not acceptable. It is important to note that there are many machine learning models and artificial intelligence techniques. Nevertheless, they should be used for solving appropriate problems. In other words, the authors proposed federated learning for estimating reference evapotranspiration, while it is not justified. Instead of federated learning, distributed learning, which is a suitable tool. can be used. Since the major contribution of this manuscript is proposinjg federated learning for estimating reference evapotranspiration, I suggest rejection.

Response:

Agreed; the contribution and justification of FL model is updated with highlighted changes on page 2 in introduction section, and on page page 5 and 6 keeping in view your concern. Hope the clear emphasis clear contribution and suggest use of FL model.

The discussion section is also updated to reflect the reasons fo not choosing the distributed approach on page 16.

Reviewer #5:

The response to your comments and suggestions are given below;

Comment 1:

The author mentions that a "feature importance-based analysis" was conducted; however, the manuscript lacks details on how the importance of the features was calculated. Please provide a clear explanation and justification for the method used to determine feature importance, including any metrics, algorithms, or criteria employed in the analysis. 

Response: 

Agreed; the description regarding the method used to calculate the Feature importance is given on page 9 and Page 10 with highlighted changes.

Comment 2:

The manuscript lacks technical details about the proposed Federated Learning-based framework for estimating ETo. Readers would benefit from more specific information about the framework's design and implementation. For example, what type of architectural design was used on the client side in FL environment for estimating ETo? How many clients participated in the training process? How many communication rounds occurred between the server and clients to exchange trained weights? Additionally, if machine learning algorithms such as Random Forest Regressor (RFR), Decision Tree Regressor (DTR), or Support Vector Regressor (SVR) were trained on the client side, it is important to explain how the trained weights were aggregated on the server side, evaluated, and then sent back to the clients. Please provide appropriate details to address these concerns, and if possible, include code to further clarify your approach.

Response:

Agreed the details of federated learning model is added on page 10,11 with highlighted changes.

We believe these revisions address your concerns and enhance the clarity of our work. Thank you once again for your valuable contributions. We look forward to your feedback on the revised manuscript.

Best regards,

Muhammad Tausif et al.

---

## [Decision Letter · Decision Letter 3]

22 Oct 2024

PONE-D-24-15636R3Federated learning based reference evapotranspiration estimation for distributed crop fieldsPLOS ONE

Dear Dr. Tausif,

Thank you for submitting your manuscript to PLOS ONE. After careful consideration, we feel that it has merit but does not fully meet PLOS ONE’s publication criteria as it currently stands. Therefore, we invite you to submit a revised version of the manuscript that addresses the points raised during the review process.

We look forward to receiving your revised manuscript.

Kind regards,

Ghani Rahman

Academic Editor

PLOS ONE

Reviewers' comments:

Reviewer's Responses to Questions

**Comments to the Author**

1. If the authors have adequately addressed your comments raised in a previous round of review and you feel that this manuscript is now acceptable for publication, you may indicate that here to bypass the “Comments to the Author” section, enter your conflict of interest statement in the “Confidential to Editor” section, and submit your "Accept" recommendation.

Reviewer #1: All comments have been addressed

Reviewer #4: (No Response)

Reviewer #5: (No Response)

2. Is the manuscript technically sound, and do the data support the conclusions?

Reviewer #1: Yes

Reviewer #4: Partly

Reviewer #5: No

3. Has the statistical analysis been performed appropriately and rigorously? 

Reviewer #1: Yes

Reviewer #4: No

Reviewer #5: (No Response)

4. Have the authors made all data underlying the findings in their manuscript fully available?

Reviewer #1: Yes

Reviewer #4: (No Response)

Reviewer #5: (No Response)

5. Is the manuscript presented in an intelligible fashion and written in standard English?

Reviewer #1: Yes

Reviewer #4: (No Response)

Reviewer #5: (No Response)

6. Review Comments to the Author

Reviewer #1: The authors of the article have satisfactorily incorporated all my comments and I have no further comments.

Reviewer #4: The justification of using federated learning seems to be improved. But not satisfactory because the results of traditional ML models rely on their input data. I recommend to modify this sentence: "Traditional machine learning methods of ETo are very effective in many contexts but limited in the scope of their applicability to specific geographical locations". When you provide traditional ML models with a local dataset, they cannot be used globally. Therefore, this is a problem related to the input data rather than ML models.

Additionally, there is no postprocessing of the ML results. Their preformances were copared only based on a few metrics. Therefore, there is a need to conduct futher analyses on the ML results, like reliability analysis or uncertainty analysis (10.1007/s40899-023-01021-y and 10.1016/j.compag.2020.105653). This is neccessary to do because you are comparing different types of ML models. Also, this can improve the discussion section and provide a better perspective of the results.

Reviewer #5: The authors' responses to the review comments lack sufficient detail, and my major points raised in the previous review have not been adequately addressed. There is a significant absence of technical details throughout the manuscript. Although the authors claim to utilize Federated Learning approaches for estimating evapotranspiration in distributed crop fields, they fail to provide specific information on how this technology is implemented within their study. The manuscript does not explain in detail how the methodology used and the steps taken to integrate Federated Learning into their analysis. Furthermore, the responses provided mainly discuss the general principles of Federated Learning rather than its application to the authors' specific research context. This generic explanation does not demonstrate how the authors' approach is uniquely tailored to their work, making it challenging to assess the validity and relevance of their findings. To enhance the manuscript, I recommend that the authors include technical descriptions of their methodology, particularly, how Federated Learning was specifically applied to their evapotranspiration estimation. This information is crucial for readers to understand and evaluate the effectiveness of the proposed approach.

7. PLOS authors have the option to publish the peer review history of their article (what does this mean?). If published, this will include your full peer review and any attached files.

Reviewer #1: No

Reviewer #4: No

Reviewer #5: No

---

## [Author Response · Author response to Decision Letter 3]

14 Nov 2024

Response to Reviewer 1’s Comments:

R1.1: The authors of the article have satisfactorily incorporated all my comments and I have no further comments.

Response: Thank you for your time and efforts to improve the manuscript.

Response to Reviewer 4’s Comments:

R4.1: The justification of using federated learning seems to be improved. But not satisfactory because the results of traditional ML models rely on their input data. I recommend to modify this sentence: ”Traditional machine learning methods of ETo are very effective in many contexts but limited in the scope of their applicability to specific geographical locations”. When you provide traditional ML models with a local dataset, they cannot be used globally. Therefore, this is a problem related to the input data rather than ML models. Additionally, there is no postprocessing of the ML results. Their preformances were copared only based on a few metrics. Therefore, there is a need to conduct futher analyses on the ML results, like reliability analysis or uncertainty analysis (10.1007/s40899-023-01021-y and 10.1016/j.compag.2020.105653). This is neccessary to do because you are comparing different types of ML models. Also, this can improve the discussion section and provide a better perspective of the results.

Response: Thank you for you comments and suggestion.

We have revised the mentioned statement as per your suggestion (Page 2, Lines 47-51).

Moreover, we perform Kruskal-Wallis to evaluate the performance of RFR, SVR and DTR and ANOVA test to analyze the reliability of the proposed approach. The results of ANOVA analysis are presented on Page 14 (Lines 488-494).

Response to Reviewer 5’s Comments:

R5.1: The authors’ responses to the review comments lack sufficient detail, and my major points raised in the previous review have not been adequately addressed. There is a significant absence of technical details throughout the manuscript. Although the authors claim to utilize Federated Learning approaches for estimating evapotranspiration in distributed crop fields, they fail to provide specific information on how this technology is implemented within their study. The manuscript does not explain in detail how the methodology used and the steps taken to integrate Federated Learning into their analysis. Furthermore, the responses provided mainly discuss the general principles of Federated Learning rather than its application to the authors’ specific research context. This generic explanation does not demonstrate how the authors’ approach is uniquely tailored to their work, making it challenging to assess the validity and relevance of their findings. To enhance the manuscript, I recommend that the authors include technical descriptions of their methodology, particularly, how Federated Learning was specifically applied to their evapotranspiration estimation. This information is crucial for readers to understand and evaluate the effectiveness of the proposed approach.

Response: Thank you for your valuable suggestion. Accordingly, we have revised the Section 3.5 (Pages 10-11, Lines 353-416).

---

## [Decision Letter · Decision Letter 4]

19 Nov 2024

Federated learning based reference evapotranspiration estimation for distributed crop fields

PONE-D-24-15636R4

Dear Dr. Tausif,

We’re pleased to inform you that your manuscript has been judged scientifically suitable for publication and will be formally accepted for publication once it meets all outstanding technical requirements.

Kind regards,

Ghani Rahman

Academic Editor

PLOS ONE

Additional Editor Comments (optional):

Reviewers' comments:

Reviewer's Responses to Questions

**Comments to the Author**

1. If the authors have adequately addressed your comments raised in a previous round of review and you feel that this manuscript is now acceptable for publication, you may indicate that here to bypass the “Comments to the Author” section, enter your conflict of interest statement in the “Confidential to Editor” section, and submit your "Accept" recommendation.

Reviewer #5: All comments have been addressed

2. Is the manuscript technically sound, and do the data support the conclusions?

Reviewer #5: Yes

3. Has the statistical analysis been performed appropriately and rigorously? 

Reviewer #5: Yes

4. Have the authors made all data underlying the findings in their manuscript fully available?

Reviewer #5: Yes

5. Is the manuscript presented in an intelligible fashion and written in standard English?

Reviewer #5: Yes

6. Review Comments to the Author

Reviewer #5: (No Response)

7. PLOS authors have the option to publish the peer review history of their article (what does this mean?). If published, this will include your full peer review and any attached files.

Reviewer #5: No

---

## [Editor Report · Acceptance letter]

17 Jan 2025

PONE-D-24-15636R4 

PLOS ONE

Dear Dr. Tausif, 

I'm pleased to inform you that your manuscript has been deemed suitable for publication in PLOS ONE. Congratulations! Your manuscript is now being handed over to our production team.

Kind regards, 

on behalf of

Dr. Ghani Rahman 

Academic Editor

PLOS ONE